# Identification of lectin receptors for conserved SARS-CoV-2 glycosylation sites

David Hoffmann[1,†], Stefan Mereiter[1,†] iD, Yoo Jin Oh[2] iD, Vanessa Monteil[3] iD, Elizabeth Elder[4] iD, Rong Zhu[2], Daniel Canena[2], Lisa Hain[2], Elisabeth Laurent[5] iD, Clemens Grünwald-Gruber[6], Miriam Klausberger[7] iD, Gustav Jonsson[1] iD, Max J Kellner[1], Maria Novatchkova[1], Melita Ticevic[1], Antoine Chabloz[8], Gerald Wirnsberger[9], Astrid Hagelkruys[1] iD, Friedrich Altmann[6], Lukas Mach[10] iD, Johannes Stadlmann[1,6], Chris Oostenbrink[11], Ali Mirazimi[3,12] iD, Peter Hinterdorfer[2] iD & Josef M Penninger[1,8,*] iD

## Abstract

New SARS-CoV-2 variants are continuously emerging with critical implications for therapies or vaccinations. The 22 *N*-glycan sites of Spike remain highly conserved among SARS-CoV-2 variants, opening an avenue for robust therapeutic intervention. Here we used a comprehensive library of mammalian carbohydrate-binding proteins (lectins) to probe critical sugar residues on the full-length trimeric Spike and the receptor binding domain (RBD) of SARS-CoV-2. Two lectins, Clec4g and CD209c, were identified to strongly bind to Spike. Clec4g and CD209c binding to Spike was dissected and visualized in real time and at single-molecule resolution using atomic force microscopy. 3D modelling showed that both lectins can bind to a glycan within the RBD-ACE2 interface and thus inter-feres with Spike binding to cell surfaces. Importantly, Clec4g and CD209c significantly reduced SARS-CoV-2 infections. These data report the first extensive map and 3D structural modelling of lectin-Spike interactions and uncovers candidate receptors involved in Spike binding and SARS-CoV-2 infections. The capacity of CLEC4G and mCD209c lectins to block SARS-CoV-2 viral entry holds promise for pan-variant therapeutic interventions.

**Keywords** glycosylation; lectin; SARS-CoV-2; spike

**Subject Categories** Immunology; Microbiology, Virology & Host Pathogen Interaction; Post-translational Modifications & Proteolysis
**The EMBO Journal (2021) 40: e108375**

## Introduction

COVID-19 caused by SARS-CoV-2 infections has triggered a pandemic massively disrupting health care, social and economic life. SARS-CoV-2 main entry route into target cells is mediated by the viral Spike protein, which binds to angiotensin converting enzyme 2 (ACE2) expressed on host cells (Monteil *et al*, 2020). The Spike protein is divided into two subunits, S1 and S2. The S1 subunits comprises the receptor binding domain (RBD) which confers ACE2 binding activity. The S2 subunit mediates virus fusion with the cell membrane following proteolytic cleavage (Hoffmann *et al*, 2020; Shang *et al*, 2020; Walls *et al*, 2020). Cryo-electron microscopy studies have shown that the Spike protein forms a highly flexible homotrimer containing 22 N-glycosylation sites each, 18 of which are conserved with the closely related SARS-CoV which caused the 2002/03 SARS epidemic (Ke *et al*, 2020; Walls *et al*, 2020). Point mutations removing glycosylation

1  IMBA, Institute of Molecular Biotechnology of the Austrian Academy of Sciences, Vienna, Austria
2  Institute of Biophysics, Johannes Kepler University Linz, Linz, Austria
3  Department of Laboratory Medicine, Unit of Clinical Microbiology, Karolinska Institute and Karolinska University Hospital, Stockholm, Sweden
4  Public Health Agency of Sweden, Solna, Sweden
5  Department of Biotechnology and BOKU Core Facility Biomolecular & Cellular Analysis, University of Natural Resources and Life Sciences, Vienna, Austria
6  Department of Chemistry, University of Natural Resources and Life Sciences, Vienna, Austria
7  Department of Biotechnology, University of Natural Resources and Life Sciences, Vienna, Austria
8  Department of Medical Genetics, Life Sciences Institute, University of British Columbia, Vancouver, BC, Canada
9  Apeiron Biologics, Vienna, Austria
10  Department of Applied Genetics and Cell Biology, University of Natural Resources and Life Sciences, Vienna, Austria
11  Department for Material Sciences and Process Engineering, Institute for Molecular Modeling and Simulation, University of Natural Resources and Life Sciences, Vienna, Austria
12  National Veterinary Institute, Uppsala, Sweden
   *Corresponding author. Tel: +43 1790 44; E-mail: josef.penninger@ubc.ca
   †These authors contributed equally to this work
   [The copyright line of this article was changed on 4 October 2021 after original online publication.]

sites of the SARS-CoV-2 Spike protein were found to yield less infectious pseudo-typed viruses (Li *et al*, 2020). As Spike and RBD glycosylation affect ACE2 binding and SARS-CoV-2 infections, targeting virus-specific glycosylation could be a novel means for therapeutic intervention.

Glycosylation of viral proteins ensures proper folding and shields antigenic viral epitopes from immune recognition (Watanabe *et al*, 2019; Watanabe *et al*, 2020b). To create this glycan shield, the virus hijacks the host glycosylation machinery and thereby ensures the presentation of self-associated glycan epitopes. Apart from shielding epitopes from antibody recognition, glycans can be ligands for lectin receptors. For instance, mannose-specific mammalian lectins, such as DC-SIGN (CD209) or its homolog L-SIGN (CD299), are well known to bind to viruses like HIV-1 and also SARS-CoV (Van Breedam *et al*, 2014). Lectin receptors are often expressed on immune and endothelial cells and serve as pattern recognition receptors involved in virus internalization and transmission (Osorio & Reis e Sousa, 2011). Recent studies have characterized the recognition of the SARS-CoV-2 Spike by previously known virus-binding lectins, such as DC-SIGN, L-SIGN, MGL, and MR (preprint: Gao *et al*, 2020). Given that SARS-CoV-2 relies less on oligo-mannose-type glycosylation, as compared to for instance HIV-1, and displays more complex-type glycosylation, it is unknown if additional lectin receptors are capable of binding the Spike protein and whether such interactions might have functional relevance in SARS-CoV-2 infections.

## Results

### Preparation of the first near genome-wide lectin library to screen for novel binders of Spike glycosylation

To systematically identify lectins that bind to the trimeric Spike protein and RBD of SARS-CoV-2, we searched for all annotated carbohydrate recognition domains (CRDs) of mouse C-type lectins, Galectins and Siglecs. Of 168 annotated CRDs, we were able to clone, express and purify 143 lectin-CRDs as IgG2a-Fc fusion proteins from human HEK293F cells (Fig 1A, Dataset EV1). The resulting dimeric lectin-Fc fusion proteins (hereafter referred to as lectins) showed a high degree of purity (Fig 1B). This collection of lectins is, to our knowledge, the first comprehensive library of mammalian CRDs.

We next recombinantly expressed monomeric RBD and stabilized full-length trimeric pre-fusion Spike protein (hereafter referred to as Spike protein) in human HEK293-6E cells. Using mass spectrometry, we characterized all 22 *N*-glycosylation sites on the full-length Spike protein and 2 *N*-glycosylation sites on the RBD (Fig 1C and Dataset EV2). Most of the identified structures were in accordance with

previous studies using full-length Spike (Watanabe *et al*, 2020a) with the exception of N331, N603 and N1194, which presented a higher structural variability of the glycan branches (Fig 1C and Dataset EV2). Importantly, the *N*-glycan sites of Spike are highly conserved among the sequenced SARS-CoV-2 viruses including the emerging variants B.1.1.7, 501Y.V2 and P.1 (Fig EV1A and B). The detected *N*-glycan species ranged from poorly processed oligo-mannose structures to highly processed multi-antennary complex *N*-glycans in a site-dependent manner. This entailed also a large variety of terminal glycan epitopes, which could act as ligands for lectins. Notably, the two glycosylation sites N331 and N343 located in the RBD carried more extended glycans, including sialylated and di-fucosylated structures, when expressed as an independent construct as opposed to the full-length Spike protein (Fig 1C and Dataset EV2). These data underline the complex glycosylation of Spike and reveal that *N*-glycosylation of the RBD within the 3D context of full-length trimeric Spike is different from *N*-glycosylation of the RBD expressed as minimal ACE2 binding domain.

### CD209c and Clec4g are novel high affinity binders of SARS-CoV-2 Spike

We evaluated the reactivity of our murine lectin library against the trimeric Spike and monomeric RBD of SARS-CoV-2 using an ELISA assay (Fig EV2A). This screen revealed that CD209c (SIGNR2), Clec4g (LSECtin) and Reg1 exhibited pronounced binding to Spike, whereas Mgl2 and Asgr1 displayed elevated binding to the RBD (Fig 2A and B, Dataset EV3). Further, we investigated the reactivity of the lectin library against human recombinant soluble ACE2 (hrsACE2); none of the lectins bound to hrsACE2 (Fig EV2B). We excluded Reg1 from further studies due to inconsistent ELISA results, likely due to protein instability. Asgr1 was excluded because it bound only to RBD but not to the Spike trimer, in accordance with the differences in glycosylation of glycosites N331 and N343 between Spike and RBD (Fig 1C). This highlights the importance of using a full-length trimeric Spike protein for functional studies. To confirm that the observed interactions were independent of protein conformation, Spike was denatured prior to the ELISA assay; binding of CD209c and Clec4g to the unfolded Spike remained unaltered (Fig 2C). Importantly, enzymatic removal of *N*-glycans by PNGase F treatment reduced the binding of CD209c, Clec4g, and Mgl2 towards Spike (Figs 2D and EV2C), confirming *N*-glycans as ligands. Binding of ACE2, which relies on protein–protein interactions, was completely abrogated when Spike was denatured (Fig 2C). These data identify lectins that have the potential to bind to the RBD and trimeric Spike of SARS-CoV-2.

Based on the robust *N*-glycan dependent Spike binding, we focused our further studies on CD209c and Clec4g. We first used surface plasmon resonance (SPR) to determine the kinetic and

**Figure 1. Lectin library and SARS-CoV-2 Spike and RBD glycosylation.**

A   Schematic overview of cloning, expression and purification of 143 carbohydrate recognition domain (CRD)—mouse IgG2a-Fc fusion proteins, from 168 annotated murine CRD containing proteins. The constructs were expressed in HEK293 cells, and secreted Fc-fusion proteins were purified using protein A columns. See Dataset EV1 for full list of expressed CRDs.
B   Exemplified SDS–PAGE of purified Clec7a and Mgl2 stained with Coomassie blue.
C   Glycosylation map of the SARS-CoV-2 Spike and RBD. The most prominent glycan structures are represented for each site, with at least 15% relative abundance. * marks highly variable glycosylation sites in which no single glycan structure accounted for > 15% relative abundance. The different monosaccharides are indicated using standardized nomenclature. NTD, n-terminal domain; RBD, receptor binding domain; S1/S2 and S2′, proteolytic cleavage sites; HR1 and HR2, α-helical heptad repeat domains 1 and 2; GlcNAc, N-acetylglucosamine.

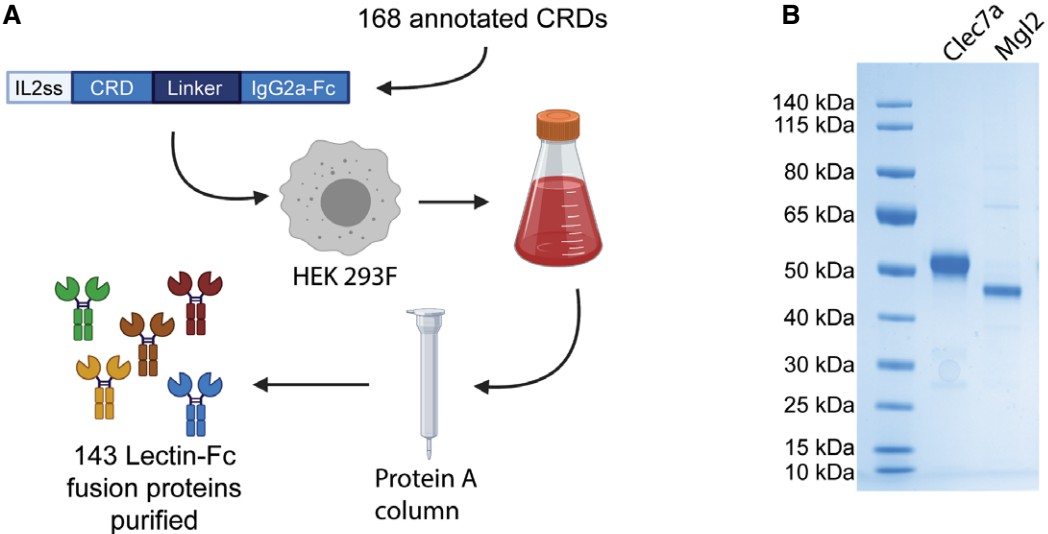

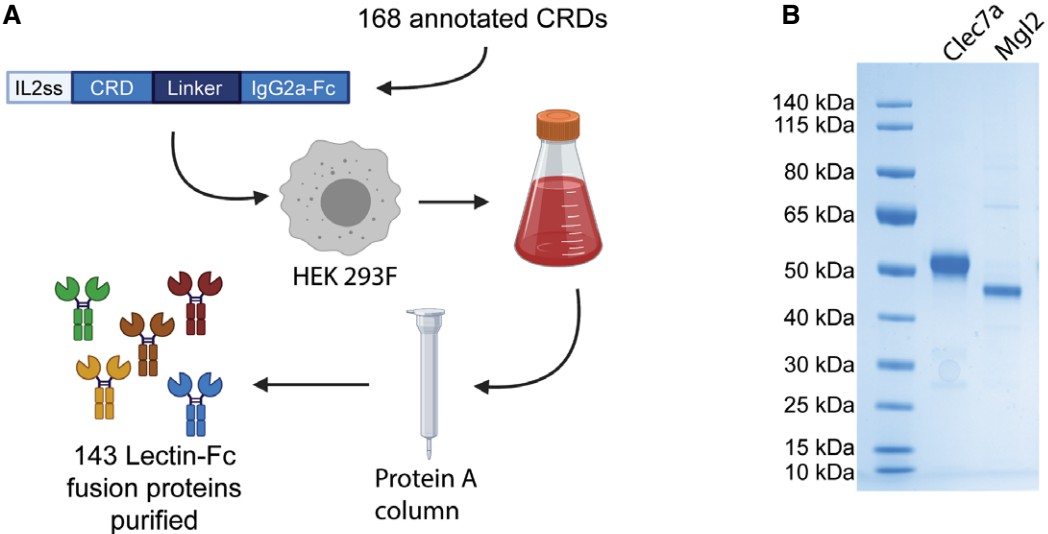

**Figure 1.**

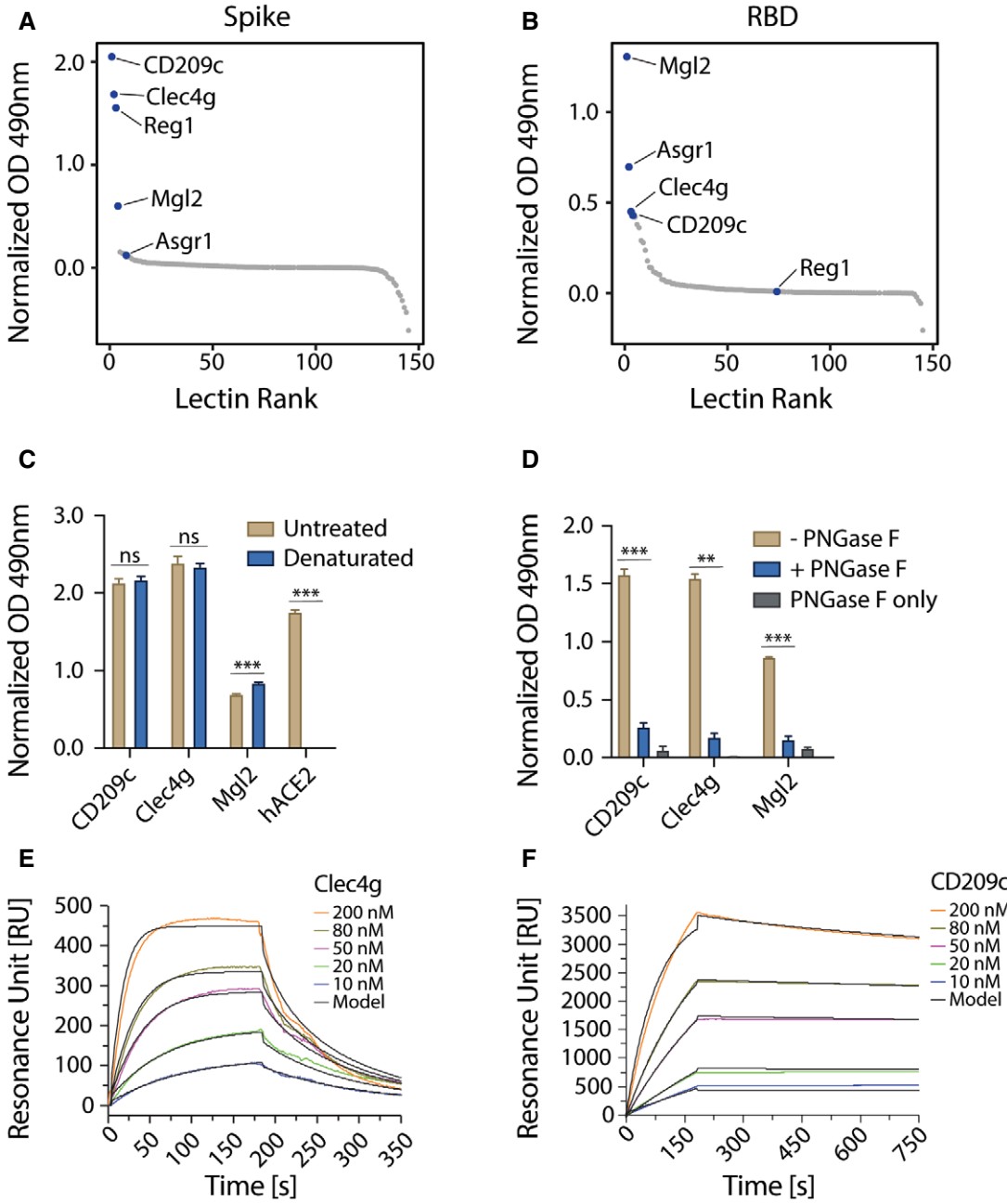

**Figure 2. Identification of lectins that bind to Spike and RBD of SARS-CoV-2.**

A, B   ELISA screen of the lectin-Fc library against full-length trimeric SARS-CoV-2 Spike (A) or monomeric RBD (B). Results are shown as mean OD values of 2 replicates normalized against a BSA control and ranked by value. Lectin-Fc fusion proteins with a normalized OD > 0.5 in either (A) or (B) are indicated in both panels. See Dataset EV2 for primary ELISA data.

C   Lectin-Fc and human ACE2-mIgG1 Fc-fusion protein (hACE2) binding to untreated or heat-denatured full-length SARS-CoV-2 Spike by ELISA. hACE2-mIgG1 was used as control for complete denaturation of Spike protein. Results are shown as mean OD values ± SD normalized to the BSA control (technical replicates, $N = 3$).

D   Lectin-Fc binding to full-length SARS-CoV-2 Spike with or without de-*N*-glycosylation by PNGase F. "PNGase F only" denotes wells that were not coated with the Spike protein. Results are shown as mean OD values ± SD normalized to BSA controls (technical replicates, $N = 3$).

E, F   Surface plasmon resonance (SPR) analysis with immobilized full-length trimeric Spike, probed with various concentrations of Clec4g-Fc (E) and CD209c-Fc (F). See Table 1 for kinetics values.

Data information: (C) *t*-test with Holm–Sidak correction for multiple comparisons. (D) One-way ANOVA with Tukey's multiple comparisons; **$P < 0.01$; ***$P < 0.001$; ns: not significant.

**Table 1.  Values computed for surface plasmon resonance (SPR) and atomic force microscopy (AFM).**

| | SPR | | | AFM | |
|---|---|---|---|---|---|
| | $k_{a,1}$ [M$^{-1}$s$^{-1}$] | $k_{d,1}$ [s$^{-1}$] | $K_{d,1}$ [M] | $k_{off}$ [s$^{-1}$] | $x_\beta$ [nm] |
| Clec4g | $6.17 \times 10^4$ | 0.0997 | $1.62 \times 10^{-6}$ | $0.037 \pm 0.007$ | $0.55 \pm 0.02$ |
| CD209c | $1.61 \times 10^4$ | 0.0159 | $0.988 \times 10^{-6}$ | $0.041 \pm 0.025$ | $0.76 \pm 0.03$ |
| hCLEC4G | $7.77 \times 10^4$ | 0.0201 | $0.259 \times 10^{-6}$ | $0.007 \pm 0.0004$ | $0.64 \pm 0.55$ |
| hCD209 | $1.32 \times 10^4$ | 0.0316 | $2.39 \times 10^{-6}$ | $0.008 \pm 0.001$ | $0.73 \pm 0.14$ |

SPR. Kinetic association ($k_{a,1}$), kinetic dissociation ($k_{d,1}$) and equilibrium dissociation ($K_{d,1}$) constants of the first binding step fitted from the bivalent analyte model, assuming two-step binding and dissociation of the lectins to adjacent immobilized Spike trimer binding sites under spontaneous thermodynamic energy barriers; no reasonable fit was obtained with the simple 1:1 binding model. AFM. Kinetic off-rate constants ($k_{off}$) and lengths of dissociation paths ($x_\beta$) of single lectin bonds, originating from force-induced unbinding in single-molecule force spectroscopy (SMFS) experiments and computed using Evans's (Bell, 1978; Evans & Ritchie, 1997) model, assuming a sharp single dissociation energy barrier.

equilibrium binding constants of these lectins to the trimeric Spike. The resulting experimental binding curves were fitted to the "bivalent analyte model" (Traxler *et al*, 2017) which assumes two-step binding of the lectin dimers to adjacent immobilized Spike trimer binding sites (Fig 2E and F). From these fits, we computed the kinetic association ($k_{a,1}$), kinetic dissociation ($k_{d,1}$) and equilibrium dissociation ($K_{d,1}$) binding constants of single lectin bonds (Table 1). The equilibrium dissociations ($K_{d,1}$) values were 1.6 µM and 1.0 µM for Clec4g and CD209c, respectively.

**Multiple CD209c and Clec4g molecules bind simultaneously to SARS-CoV-2 Spike and form compact complexes**

To study Spike binding of these two lectins at the single-molecule level, we used atomic force microscopy (AFM) and performed single-molecule force spectroscopy (SMFS) experiments. To this end, we coupled trimeric Spike to the tip of the AFM cantilever and performed single-molecule force measurements (Hinterdorfer *et al*, 1996), by moving the Spike trimer-coupled tip towards the surface-bound lectins to allow for bond formations (Fig 3A). Unbinding was accomplished by pulling on the bonds, which resulted in characteristic downward deflection signals of the cantilever, whenever a bond was ruptured (Fig 3B). The magnitude of these vertical jumps reflects the unbinding forces, which were of typical strengths for specific molecular interactions (Rankl *et al*, 2008). Using this method (Rankl *et al*, 2008; Zhu *et al*, 2010), we quantified unbinding forces (Fig 3C) and calculated the binding probability and the number of bond ruptures between CD209c or Clec4g and trimeric

Spike (Fig 3D). Both lectins showed a very high binding probability and could establish up to 3 strong bonds with accumulating interaction force strengths reaching 150 pN in total with trimeric Spike (Fig 3C), with the preference of single and dual bonds (Figs 3C and D, and EV3A and B, Table 1). Of note, multi-bond formation leads to stable complex formation, in which the number of formed bonds enhances the overall interaction strength and dynamic stability of the complexes. To assess dynamic interactions between single molecules of trimeric Spike and the lectins in real time we used high-speed AFM (Kodera *et al*, 2010; Preiner *et al*, 2014). Addition of Clec4g and CD209c led to a volume increase of the lectin/Spike complex in comparison with the trimeric Spike alone; based on the volumes we could calculate that on average 3.2 molecules of Clec4g and 5.2 molecules of CD209c were bound to one Spike trimer (Figs 3E and EV3C–F and Movies EV1–EV4). These data show, in real-time, at single-molecule resolution, that mouse Clec4g and CD209c can directly associate with trimeric Spike.

**The human lectins CD209 and CLEC4G are high affinity receptors for SARS-CoV-2 Spike**

Having characterized binding of murine CD209 and Clec4g to Spike, we next assessed whether their closest human homologues, namely, human CD209 (hCD209), and human CD299 (hCD299) for murine CD209c, and human CLEC4G (hCLEC4G) for mouse Clec4g, can also bind to full-length trimeric Spike of SARS-CoV-2. hCD209, hCD299 and hCLEC4G indeed exhibited binding to Spike (Fig 4A), demonstrating conserved substrate specificities. The binding of these

**Figure 3.  Single molecule, real-time imaging of lectin-Spike binding.**

A   Schematic overview of single-molecule force spectroscopy (SMFS) experiments using full-length trimeric Spike coupled to an atomic force microscopy (AFM) cantilever tip and surface coated murine Clec4g-Fc or CD209c-Fc. Arrow indicates pulling of cantilever.

B   Representative force traces showing sequential bond ruptures in the SMFS experiments. Measured forces are shown in pico-Newtons (pN).

C   Experimental probability density function (PDF) of unbinding forces (in pN) determined by SMFS (black line, measured data). The three distinct maxima fitted by a multi-Gaussian function reveal rupture of a single bond (blue dotted line), or simultaneous rupture of 2 (red dotted line) and 3 (green dotted line) bonds, respectively.

D   SMFS-determined binding probability for the binding of trimeric Spike to Clec4g and CD209c. Data are shown as mean binding probability ± SD of single, double, triple or quadruple bonds (technical replicates, N = 4).

E   High-speed AFM of single trimeric Spike visualizing the real-time interaction dynamics with lectins. Top panel shows 5 frames of trimeric Spike alone imaged on mica. Middle and bottom panels show 5 sequential frames of trimeric Spike/Clec4g and trimeric Spike/CD209c complexes, acquired at a rate of 153.6 and 303 ms/frame, respectively. Association and dissociation events between lectin and Spike are indicated by white and red arrows, respectively. The blue dotted ellipses display the core of the complexes showing low conformational mobility. Colour schemes indicate height of the molecules in nanometers (nm). Volumes of single trimeric Spike, trimeric Spike/Clec4g and trimeric Spike/CD209c complexes are indicated, as well as numbers of lectins bound to trimeric Spike, averaged over the experimental recording period.

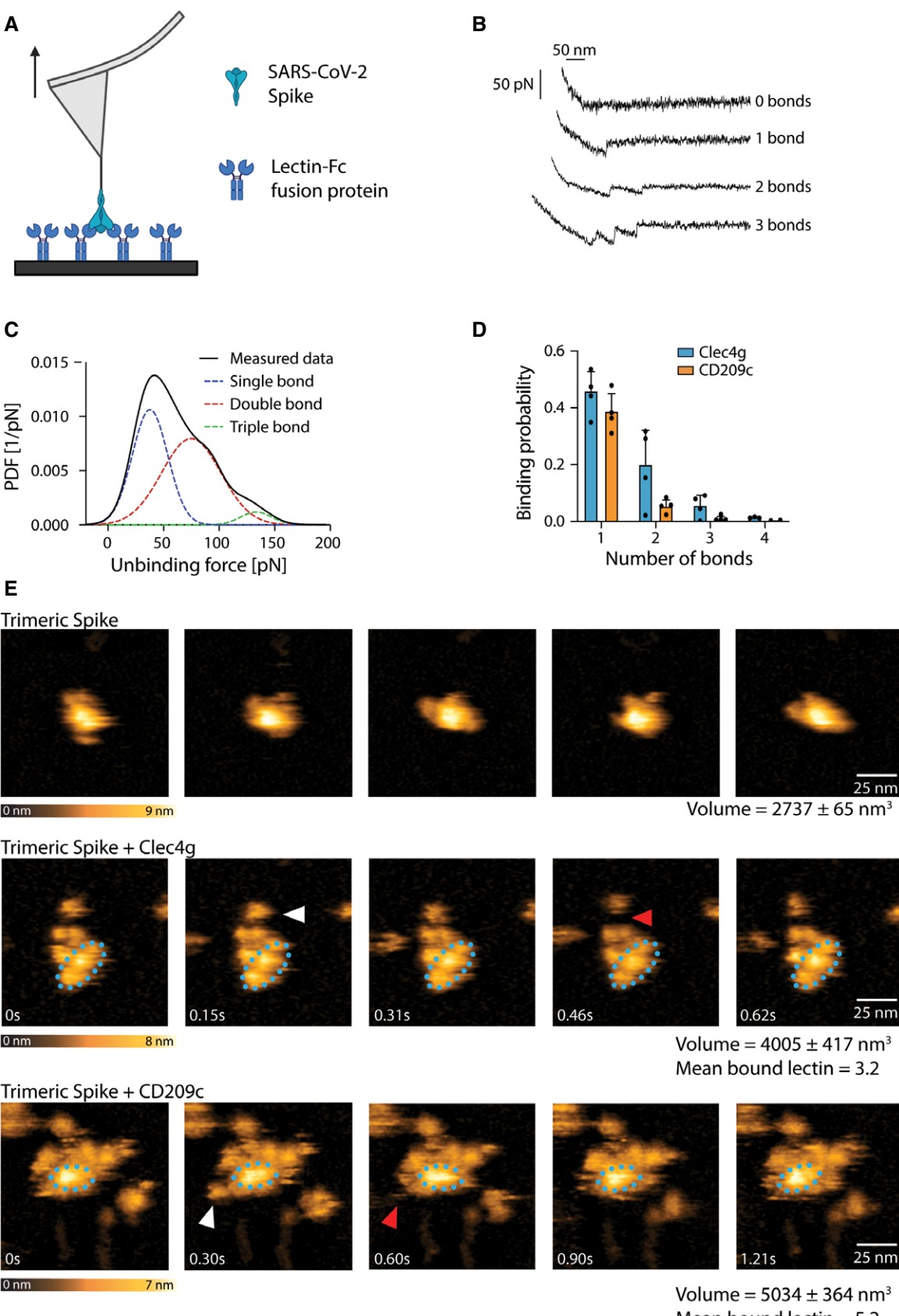

**A**

SARS-CoV-2 Spike

Lectin-Fc fusion protein

**B**

50 nm

50 pN

0 bonds

1 bond

2 bonds

3 bonds

**C**

— Measured data
--- Single bond
--- Double bond
--- Triple bond

**D**

■ Clec4g
■ CD209c

**E**

Trimeric Spike

0 nm          9 nm

Volume = 2737 ± 65 nm³

Trimeric Spike + Clec4g

0s          0.15s          0.31s          0.46s          0.62s          25 nm

0 nm          8 nm

Volume = 4005 ± 417 nm³
Mean bound lectin = 3.2

Trimeric Spike + CD209c

0s          0.30s          0.60s          0.90s          1.21s          25 nm

0 nm          7 nm

Volume = 5034 ± 364 nm³
Mean bound lectin = 5.2

Figure 3.

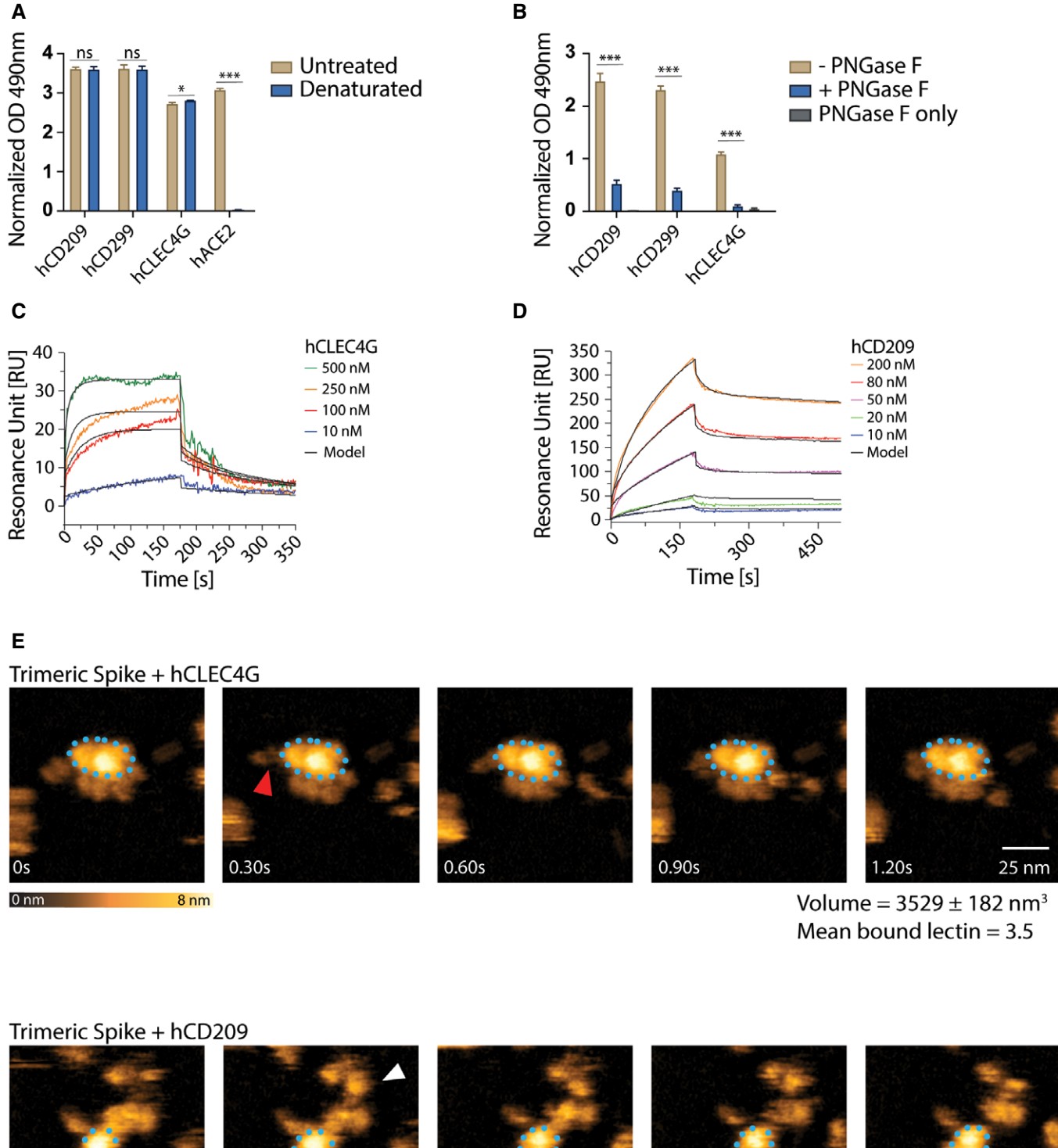

**Figure 4.**

**Figure 4.   Characterization of human lectin-Spike interactions.**

A    ELISA analyses of human lectin-hIgG1 Fc-fusion protein (hCLEC4g, hCD209, hCD299) binding to untreated or heat-denatured full-length SARS-CoV-2 Spike. A human ACE2-hIgG1-Fc fusion protein (hACE2) was used to control for the complete denaturation of Spike. Results are shown as mean OD values $\pm$ SD normalized to a BSA control (technical replicates, $N = 3$).

B    hCLEC4g, hCD209 and hCD299 binding to full-length SARS-CoV-2 Spike with or without de-*N*-glycosylation by PNGase F. "PNGase F only" denotes wells that were not coated with the Spike protein. Results are shown as mean OD values $\pm$ SD normalized against the BSA control (technical replicates, $N = 3$).

C, D    SPR analysis with immobilized full-length trimeric Spike, probed with various concentrations of hCLEC4G (C) and hCD209 (D). See Table 1 for kinetics values.

E    High-speed AFM of single trimeric Spike visualizing the real-time interaction dynamics with lectins. Top and bottom panels show 5 sequential frames of trimeric Spike/hCLEC4G and trimeric Spike/hCD209 complexes, acquired at a rate of 303 ms/frame. Association and dissociation events between hCLEC4G or hCD209 and Spike are indicated by white and red arrows, respectively. The blue dotted ellipses display the core of the complexes showing low conformational mobility. Colour schemes indicate height of the molecules in nanometers (nm). Volumes of single trimeric Spike and trimeric Spike/Clec4g and trimeric Spike/CD209c complexes are indicated, as well as numbers of lectins bound to trimeric Spike, averaged over the experimental recording period.

Data information: (A) *t*-test with Holm–Sidak correction for multiple comparisons. (B) One-way ANOVA with Tukey's multiple comparisons; *$P < 0.05$; ***$P < 0.001$; ns: not significant.

human lectins was again independent of Spike folding and abrogated by *N*-glycan removal (Fig 4A and B). SPR measurements of hCLEC4G and hCD209 bond formation to Spike showed equilibrium dissociation ($K_{d,1}$) values of 0.3 and 2.4 µM, respectively, in which the high affinity of hCLEC4G is mainly contributed by its rapid kinetic association rate constant (Fig 4C and D and Table 1). In addition, hCD209 and hCLEC4G showed a high binding probability by AFM with the formation of up to 3 bonds per trimeric Spike (Fig EV4A–C). When we monitored the dynamic interactions of the lectins with the Spike using high-speed AFM, we observed binding of—on average—3.5 hCLEC4G and 3.6 hCD209 molecules per trimeric Spike (Figs 4E and EV4D–G). In summary, our data using ELISA, SMFS, surface plasmon resonance, and high-speed atomic force microscopy show that the human lectins CLEC4G and CD209 can bind to trimeric Spike of SARS-CoV-2, in which the overall interaction strength and dynamic stability leads to compact complex formation.

**CLEC4G sterically interferes with Spike/ACE2 interaction**

We next 3D modelled binding of hCLEC4G and hCD209 to the candidate glycosylation sites present on Spike and how such attachment might relate to the binding of the trimeric Spike protein to its receptor ACE2. hCD209 is known to bind with high affinity to oligo-mannose structures (Guo *et al*, 2004). The N234 glycosylation site is the only site within the Spike that carries exclusively oligo-mannose glycans with up to 9 mannose residues (Fig 1C, Dataset EV2). 3D modelling revealed that the oligo-mannose glycans on N234 are accessible for hCD209 binding on all 3 monomers comprising the trimeric Spike (Fig 5 and Appendix Fig S1A and B). Superimposition of ACE2 interacting with the RBD and hCD209 binding to N234, showed that the hCD209 binding occurs at the lateral interface of Spike, distant from the RBD (Fig 5).

Human CLEC4G and mouse Clec4g, on the other hand, were shown to have a high affinity for complex *N*-glycans terminating with GlcNAc (Powlesland *et al*, 2008; Pipirou *et al*, 2011). To further assess the detailed ligand specificity of these two lectins, we performed glycan microarray analyses comprising of 144 different glycan structures. Our analysis revealed a remarkable specificity of both, human CLEC4G and mouse Clec4g, exclusively for N-glycans with an unsubstituted GlcNAcβ-1,2Manα-1,3Man arm (Fig 5A, Appendix Fig S2A and B, Appendix Fig S3 and Dataset EV4). In addition, our analysis revealed that mouse CD209c exhibits overlapping ligand specificities with murine Clec4g and human CLEC4G, which is in fact distinct from the known binding profile of human CD209. However, unlike Clec4g, CD209c recognized all *N*-glycan structures that displayed terminal unsubstituted GlcNAc residues, independently of the position in the glycan antennae (Fig 5A, Appendix Fig S2C, Appendix Fig S3 and Dataset EV4). The *N*-glycan at N343, located within the RBD, is the glycosylation site most abundantly decorated with terminal GlcNAc in Spike (Fig 1C, Dataset EV2), constituting the candidate binding site for murine CD209c and murine and human CLEC4G.

The terminal GlcNAc glycans on position N343 are accessible for hCLEC4G binding on all 3 Spike monomers, but in contrast to hCD209, hCLEC4G binding interferes with the ACE2/RBD interaction (Fig 5B, Appendix Fig S1C and D). Since murine Clec4g and murine CD209c show strongly overlapping ligand specificities to hCLEC4G, we next modelled binding of these two lectins to the N343 glycan site. As predicted from our data, both murine Clec4g and murine CD209c indeed interfere with the ACE2/RBD interaction (Appendix Fig S1E). Thus, whereas hCD209 is not predicted to directly affect ACE2/RBD binding, murine Clec4g, murine CD209c and human CLEC4G binding to the N343 glycan impedes Spike binding to ACE2.

**Figure 5.   Glycan ligand characterization and structural modelling.**

A    Glycan microarray analyses of mClec4g, mCD209c and hCLEC4G. Terminal GlcNAc, Gal or Man are colour-coded along the X-axis. Miscellaneous epitopes could not unambiguously be grouped to either terminal GlcNAc, Gal or Man. Mucin type O-glycans, characterized by a reducing-end GalNAc, are highlighted. The identified main binding epitope of each lectin is shown in red. Data represent the average and standard deviation of three independent technical replicates.

B    3D structural modelling of glycosylated trimeric Spike (green with glycans in yellow) interacting with glycosylated human ACE2 (purple with glycans in salmon). The CRD of hCLEC4G (cyan with Ca$^{2+}$ in orange) was modelled onto Spike monomer 3 glycan site N343 (complex-type glycan with terminal GlcNAc in purple-blue), and the CRD of hCD209 (dark blue with Ca$^{2+}$ in orange) was modelled onto the Spike monomer 3 glycan site N234 (Oligomannose structure Man9 in red). Structural superposition of CLEC4G and ACE2 highlights sterical incompatibility.

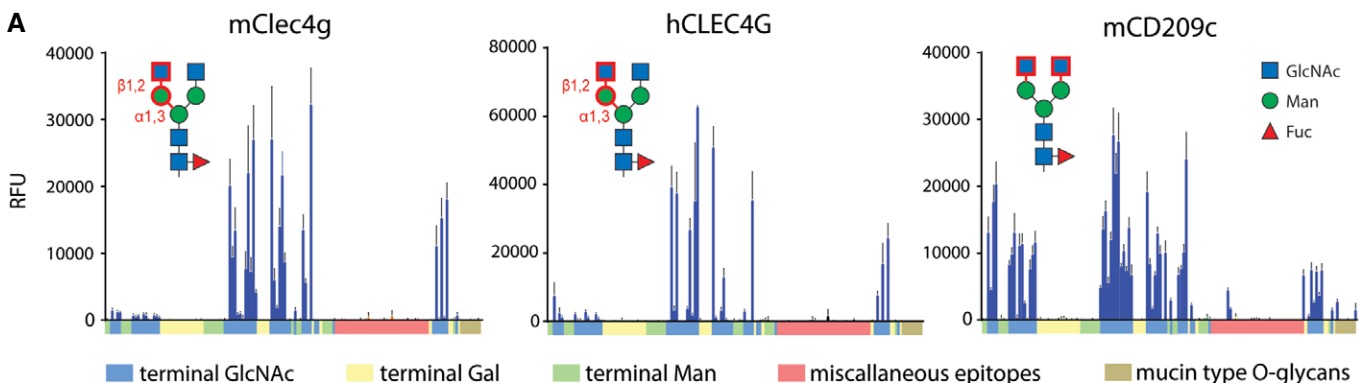

**Figure 5.**

## CD209c and CLEC4G block SARS-CoV-2 infection

To test our structural models experimentally, we assessed whether these lectins could interfere with Spike binding to the surface of Vero E6 cells, a frequently used SARS-CoV-2 infection model (Monteil *et al*, 2020). To determine this, we set-up an AFM based method, measuring spike binding activity on Vero E6 cells. Strikingly, as predicted by the structural modelling, we

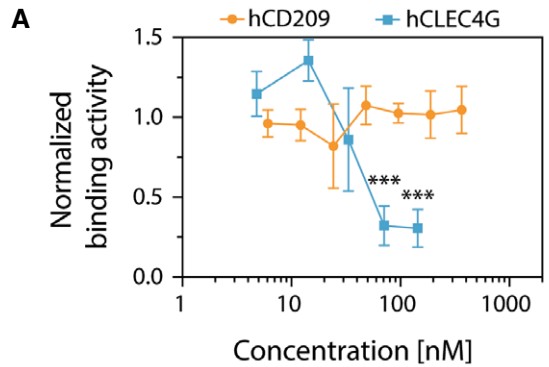

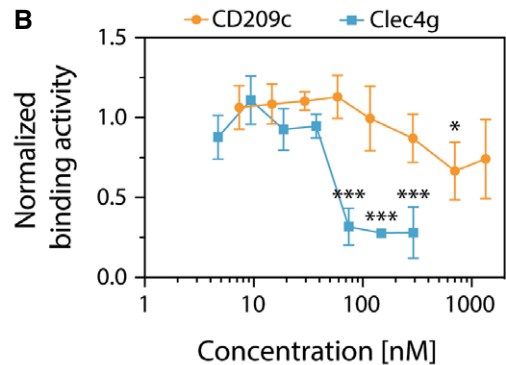

**C**  VeroE6 cells; MOI 0.02 (10³ PFU)

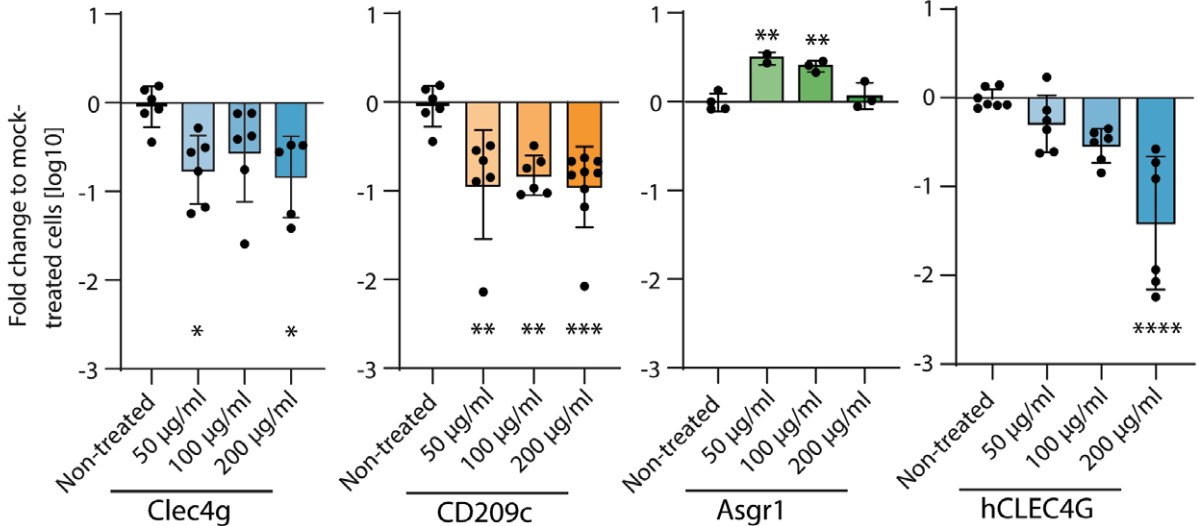

**D**  Calu3 cells; MOI 0.02 (10³ PFU)

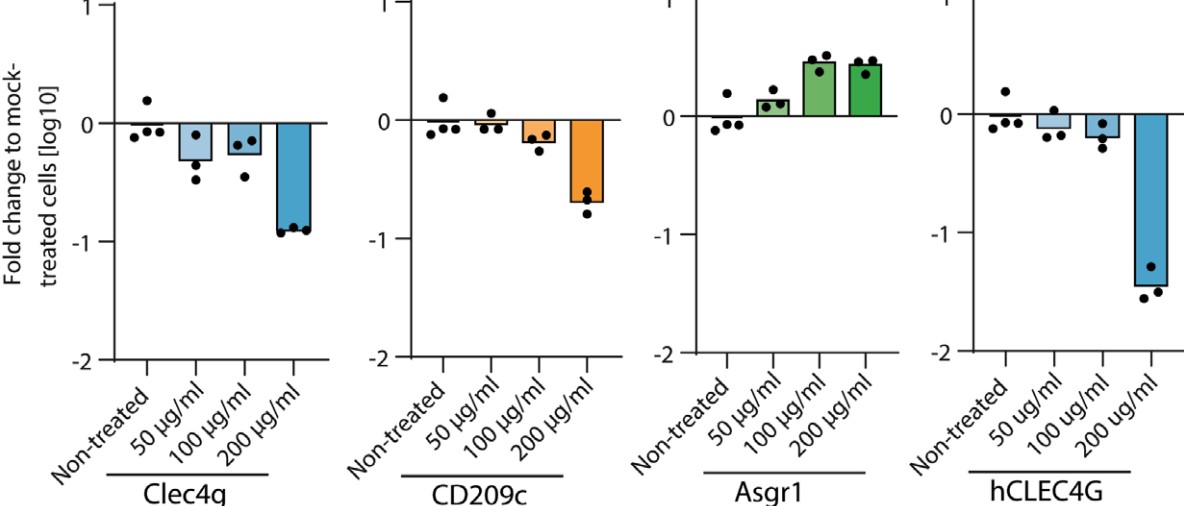

Figure 6.

**Figure 6. Functional determination of lectin-Spike binding in SARS-CoV-2 infections.**

A, B   Binding activity of the full-length trimeric Spike coupled to an AFM cantilever tip to the surface of Vero E6 cells in the presence of (A) hCLEC4G and hCD209 or (B) mouse Clec4g or murine CD209c, probed at the indicated concentrations in SMFS experiments. A definite decrease in binding was observed at concentrations of 37–75 nM for Clec4g and 33–71 nM for hCLEC4G. Data are normalized to the untreated control and shown as mean $\pm$ SD (biological replicates, $N = 4$).

C   Infectivity of mouse Clec4g, CD209c, Asgr1 and hCLEC4G pre-treated SARS-CoV-2 virus in Vero E6 cells. Data represent two to three independent experiments for Clec4g, CD209c and hCLEC4G lectins ($N = 5–9$) and, one experiment for Asgr1 ($N = 2–4$).

D   Infectivity of mouse Clec4g, CD209c, Asgr1 and hCLEC4G pre-treated SARS-CoV-2 virus in human airway epithelial Calu3 cells measured in triplicates and compared to non-treated SARS-CoV-2-infected Calu3 cells measured in quadruplicates (technical replicates).

Data information: (C) and (D) Viral RNA was measured with qRT–PCR 15 h after infection with 0.02 MOI ($10^3$ PFUs) of SARS-CoV-2. Data are presented as fold changes of viral loads over mock (only SARS-CoV-2 was added) treated controls (mean log10 values $\pm$ SD). All no virus controls were negative for SARS-CoV-2. (A-C) ANOVA with Dunnett's multiple comparisons test with the mock-treated group; *$P < 0.05$; **$P < 0.01$; ***$P < 0.001$, ****$P < 0.0001$.

found that hCLEC4G, but not hCD209, significantly interfered with the binding of trimeric Spike to the Vero E6 cell surface (Fig 6A). Similarly, mouse Clec4g, and importantly also murine CD209c, albeit to a lesser extent, interfered with Spike binding to Vero E6 cells (Fig 6B). Moreover, using competition ELISA assays, the lectins mClec4g and hCLEC4G, were able to outcompete binding of hrsACE2 to full-length trimeric Spike (Fig EV5A–C). Finally, we tested the ability of these lectins to reduce the infectivity of SARS-CoV-2. In accordance with our model, murine Clec4g, murine CD209c and human CLEC4G, but not murine Asgr1, significantly reduced SARS-CoV-2 infection of Vero E6 cells (Fig 6C). Of note, the lectin Asgr1 showed in our ELISA screen strong binding to the RBD but not to full-length trimeric Spike (Fig 2A and B). This highlights the importance of trimeric Spike rather than isolated RBD domains when screening for lectin binding and underlines the capacity of our screening strategy to identify potent inhibitory lectins of SARS-CoV-2. Finally, we tested the ability of the lectins to prevent infection of the human airway cell line Calu3. In line with our model and other data, murine Clec4g, murine CD209c and hCLEC4G significantly reduced SARS-CoV-2 infection of these human airway cells (Fig 6D). These data show that the lectins CLEC4G and CD209c can interfere with SARS-CoV-2 infections.

## Discussion

Here we report the results from unbiased screening of a comprehensive mammalian lectin library for potent SARS-CoV-2 Spike binding, identifying mouse CD209c and Clec4g, as well as their human homologs hCD209 and hCLEC4G, as N-glycan dependent Spike receptors. hCD209 has been identified as candidate receptor for SARS-CoV-2 by other groups, and other lectins have been also implicated in cellular interactions with Spike (preprint: Gao *et al*, 2020; Thépaut *et al*, 2021). CLEC4G has been reported to associate with SARS-CoV (Gramberg *et al*, 2005). High-speed atomic force microscopy allowed us to directly observe Spike/lectin interactions in real time. Our high-speed AFM data showed that Spike/CLEC4G form more rigid clusters with lower conformational flexibility as compared to the Spike/CD209 complexes in equilibrium conditions. This is in accordance with faster association rates and shorter dissociation paths of CLEC4G as compared to CD209. The experimentally observed association of 3–4 CLEC4G molecules with one molecule of the trimeric spike indicates the formation of high affinity bonds to 1–2 glycosylation sites per monomeric subunit of the trimeric Spike.

Since glycosylation is not a template driven process, but rather depends on the coordinated action of many glycosyltransferases and glycosidases (Stanley *et al*, 2015), each glycosylation site can —within some boundaries—carry a range of glycans. As a consequence, the 3 monomers of Spike can harbour different glycans on the same glycosylation site. The full-length Spike protein applied in this study is a stabilized pre-fusion Spike and carries two proline mutations and a mutated furin cleavage site. Our glycoproteomic data showed that our recombinantly expressed Spike has a remarkably similar glycan profile as reported for Spike from infectious SARS-CoV-2 produced in human airway cells (Brun *et al*, 2021). We identified N343 as the one glycosylation site that is almost exclusively covered with GlcNAc terminated glycans, the ligands of human CLEC4G and mouse Clec4g as well as, based on our new data, mouse CD209c. Given localization of N343 within the RBD, we hypothesized that CLEC4G and mCD209c binding interferes with the RBD-ACE2 interaction. Indeed, we found that murine Clec4g and human CLEC4G can functionally impede Spike binding to host cell membranes, thereby providing a rationale how this lectin can affect SARS-CoV-2 infections. Due to the dimeric structure of the lectin constructs used in our study, they act as bivalent effective inhibitors with $K_I$ values (~35–70 nM) that are higher than the affinity constants derived by SPR for single lectin bonds (~μM). Of note, in multiple attempts to produce full-length Spike harbouring the N343Q point mutation we observed that this is a crucial site for Spike stability and secretion (Appendix Fig S4). This puts in question previous conclusions drawn on Spike N343Q (Li *et al*, 2020; Lu *et al*, 2021); further studies are required to fully understand the role of specific Spike glycan sites for stability and function.

As for murine CD209c, our atomic force microscopy data indicate that CD209c engages in a larger number of interactions per trimeric Spike (~5.2), presumably because mCD209c binds to a greater variety of GlcNAc-linkages than Clec4g. This more promiscuous glycan binding of mCD209c might also explain less, albeit significant, inhibition of Spike binding to VeroE6 surface as compared to human and mouse CLEC4G. Interestingly, the efficiency of inhibiting SARS-CoV-2 infection between murine and human CLEC4G and mCD209c remained comparable. As such, the multi-valent binding of CD209c may block SARS-CoV-2 infection not only through direct interference with ACE2 binding, but possibly also via affecting conformational Spike structures or proteolytic cleavage. Of note, while human and mouse CLEC4G as well as mouse CD209c can interfere with RBD-ACE2 binding, human CD209 does apparently not associate with glycans near the RBD and hence does not block Spike binding to cells. This is in agreement to its proposed high affinity

oligo-mannosidic ligands, presented at N234, which is localized at a distance to the RBD-ACE2 interface.

Lectins play critical roles in multiple aspects of biology such as immune responses, vascular functions, or as endogenous receptors for various human pathogens. Hence, our library containing 143 lectins will now allow to comprehensively probe and map glycan structures on viruses, bacteria or fungi, as well as during development or on cancer cells, providing novel insights on the role of lectin–glycosylation interactions in infections, basic biology and disease. For instance, CD209 is expressed by antigen presenting dendritic cells, as well as inflammatory macrophages (Garcia-Vallejo & van Kooyk, 2013) and is known to bind to a variety of pathogens, such as HIV and Ebola, but also Mycobacterium tuberculosis or Candida albicans (Appelmelk *et al*, 2003). CLEC4G is strongly expressed in liver and lymph node sinusoidal endothelial cells and can also be found on stimulated dendritic cells and macrophages (Dominguez-Soto *et al*, 2007; Dominguez-Soto *et al*, 2009). CD299, one of the two homologues of mouse CD209c, which we also identified to possess Spike binding ability, is co-expressed with CLEC4G on liver and lymph node sinusoidal endothelial cells (Liu *et al*, 2004). Sinusoidal endothelial cells are important in the innate immune response, by acting as scavengers for pathogens and antigen cross-presenting cells (Knolle & Wohlleber, 2016). Thus, lectin binding to Spike might couple SARS-CoV-2 infections to antiviral immunity, which needs to be further explored. Since viral protein glycosylation depends on the glycosylation machineries of the infected cells which assemble viral particles (Watanabe *et al*, 2019), slight changes in glycosylation might explain differences in antiviral immunity and possibly severity of the disease, with critical implications for vaccine designs. Moreover, Spike binding lectins could enhance viral entry in tissues with low ACE2 expression, thus extending the organ tropism of SARS-CoV-2.

Intriguingly, *N*-glycan sites of Spike remain highly conserved among SARS-CoV-2 variants, indicating a selection pressure to preserve these sites. Thus, the capacity of CLEC4G and mCD209c lectins to block SARS-CoV-2 viral entry holds promise for pan-variant therapeutic interventions.

# Materials and Methods

### Identification of proteins containing carbohydrate recognition domains (CRDs)

Mouse lectin sequences were obtained using a domain-based approach. Briefly, proteins with a C-type lectin-like/IPR001304 domain were downloaded from InterPro 66.0 and supplemented with proteins obtained in jackhmmer searches using the PF00059.20 lectin C-type domain definition versus the mouse-specific UniProt and Ensembl databases. The collected set of candidate mouse lectins was made non-redundant using nrdb 3.0. The C-type lectin-like regions were extracted from the full-length proteins using the SMART CLECT domain definition with hmmsearch v3.1b2 and extended by 5 amino acids on both sides. To reduce redundancy, principal isoforms were selected using appris 2016_10.v24. In addition, the CRD domains for Galectins and Siglecs were added. In case a gene contained more than 1 CRD, all CRDs were cloned separately and differentiated by _1, _2, etc.

### Cloning of C-type lectin expression vectors

We used the pCAGG_00_ccb plasmid and removed the toxic ccb element by cleaving the plasmid with BsaI. Thereafter, we inserted a Fc-fusion construct, consisting of the IL2 secretion signal, followed by an EcoRV restriction site, a (GGGS)$_3$ linker domain and the mouse IgG2a-Fc domain. Subsequently, each identified CRD, was cloned in-frame into the EcoRV site.

### Transfection and purification of the lectin-mIgG2a fusion proteins

The CRD containing plasmids were transfected into Freestyle™ 293-F cells. Briefly, the day before transfection, 293-F cells were diluted to $0.7 \times 10^6$ cells/ml in 30 ml Freestyle™ 293-F medium and grown at 120 rpm at 37°C with 8% $CO_2$. The next morning, 2 µl polyethylenimine (PEI) 25K (1 mg/ml; Polysciences, 23966-1) per µg of plasmid DNA were mixed with pre-warmed Opti-MEM media (Thermo Fisher Scientific, 31985-062) to a final volume of 950 µl in tube A. In tube B, 1 µg of DNA per ml of media was mixed with pre-warmed Opti-MEM to a final volume of 950 µl. Then, the contents of tube A and B were mixed, vortexed for 1 min and incubated at room temperature for 15 min. Thereafter, the transfection mixture was added to the cell suspension. 24 h after transfection, EX-CELL 293 Serum-Free Medium (Sigma-Aldrich, 14571C) was added to a final concentration of 20%. The transfected cells were grown for 120 h and the supernatants, containing the secreted lectin-mIgG2a fusion proteins, harvested by centrifugation at 250 *g* for 10 min.

Purification of the mouse lectin-mIgG2a fusion proteins was performed using Protein A agarose resin (Gold Biotechnology, P-400-5). The protein A beads were pelleted at 150 *g* for 5 min and washed once with 1× binding buffer (0.02 M Sodium Phosphate, 0.02% sodium azide, pH = 7.0), before resuspending in 1× binding buffer. Immediately preceding purification, aggregates were pelleted from the cell culture supernatant by centrifugation for 10 min at 3,000 *g*. 10× binding buffer was added to the cell culture supernatant to a final concentration of 1× as well as 4 µl of protein A beads per ml of cell culture supernatant. The bead/ supernatant mixture was incubated overnight at 4°C. The next morning, beads were collected by centrifugation for 5 min at 150 *g*, washed twice with 20 and 10 bead volumes of 1× binding buffer, the bead pellets transferred to a 1 ml spin column (G-Biosciences, 786-811) and washed once more with 1 bead volume of 1× binding buffer. Excess buffer was removed by centrifugation at 100 *g* for 5 s. Lectin-mIgG2a fusion proteins were eluted from the protein A beads by resuspending the beads in 1 bead volume of Elution buffer (100 mM Glycine-HCl, pH = 2–3, 0.02% sodium azide). After 30 s of incubation, the elution buffer was collected into a 2 ml Eppendorf tube, containing Neutralization buffer (1 M Tris, pH = 9.0, 0.02% sodium azide) by centrifugation at 100 *g* for 15 s. Elution was performed for a total of three times. The three eluted fractions were pooled and the protein concentrations measured with the Pierce™ BCA Protein Assay Kit (Thermo Fisher Scientific, 23225) using the Pierce™ Bovine Gamma Globulin Standard (Thermo Fisher Scientific, 23212). To confirm the purity of the eluted lectin-IgG fusion proteins, we performed an SDS–PAGE, followed by a Coomassie staining. Briefly, 1 µg of eluted lectin-IgG fusion protein was mixed with Sample Buffer, Laemmli 2× Concentrate (Sigma-Aldrich,

S3401) and heated to 95°C for 5 min. Thereafter, the samples were loaded onto NuPage™ 4–12% Bis-Tris gels (Thermo Fisher Scientific, NP0321BOX) and run in 1× MOPS buffer at 140 V for 45 min. The gel was subsequently stained with InstantBlue™ Safe Coomassie Stain (Sigma-Aldrich, ISB1L) for 1 h and de-stained with distilled water. The gel picture was acquired with the ChemiDoc MP Imaging System (Bio-Rad) in the Coomassie setting.

## Recombinant expression of SARS-CoV-2 Spike protein and the receptor binding domain (RBD)

Recombinant protein expression was performed by transient transfection of HEK293-6E cells, licensed from National Research Council (NRC) of Canada, as previously described (Durocher *et al*, 2002; Lobner *et al*, 2017). Briefly, HEK293-6E cells were cultivated in FreeStyle F17 expression medium (Thermo Fisher Scientific, A1383502) supplemented with 0.1% (v/v) Pluronic F-68 (Thermo Fisher Scientific, 24040032) and 4 mM L-glutamine (Thermo Fisher Scientific, 25030081) in shaking flasks at 37°C, 8% $CO_2$, 80% humidity and 130 rpm in a Climo-Shaker ISF1-XC (Adolf Kühner AG). The pCAGGS vector constructs, containing either the sequence of the SARS-CoV-2 RBD (residues R319-F541) or the complete luminal domain of the Spike protein (modified by removing all arginine (R) residues from the polybasic furin cleavage site RRAR and introduction of the stabilizing point mutations K986P and V987P) were kindly provided by Florian Krammer, Icahn School of Medicine at Mount Sinai (NY, United States) (Amanat *et al*, 2020; Stadlbauer *et al*, 2020). High-quality plasmid preparations for transfection were kindly provided by Rainer Hahn and Gerald Striedner (University of Natural Resources and Life Sciences, Vienna, Austria). Transient transfection of the cells was performed at a cell density of approximately $1.7 \times 10^6$ cells/ml culture volume using a total of 1 μg of plasmid DNA and 2 μg of linear 40-kDa PEI (Polysciences, 24765-1) per ml culture volume. 48 and 96 h after transfection, cells were supplemented with 0.5% (w/v) tryptone N1 (Organotechnie 19553) and 0.25% (w/v) D(+)-glucose (Carl Roth X997.1). Soluble proteins were harvested after 120–144 h by centrifugation (10,000 *g*, 15 min, 4°C).

## SARS-CoV-2 Spike mutation frequency data

Annotated SARS-CoV-2 Spike mutations were extracted from the virus repository nextstrain (https://nextstrain.org/); data are from sequences deposited until March 14th, 2021. The sites of N and S/T of all 22 N-glycan sequons (N-X-S/T) were extracted, statistically compared using two-tailed Student's *t*-test and plotted with GraphPad Prism.

## Purification of recombinant trimeric Spike protein and monomeric RBD of SARS-CoV-2

For purification, the supernatants were filtered through 0.45 μm membrane filters (Merck Millipore HAWP04700), concentrated and diafiltrated against 20 mM phosphate buffer containing 500 mM NaCl and 20 mM imidazole (pH 7.4) using a Labscale TFF system equipped with a 5 kDa cut-off Pellicon™ XL device (Merck Millipore, PXC005C50). His-tagged trimer Spike and monomeric RBD were captured using a 5 ml HisTrap FF crude column (Cytiva,

17528601) connected to an ÄKTA pure chromatography system (Cytiva). Bound proteins were eluted by applying a linear gradient of 20–500 mM imidazole over 20 column volumes. Fractions containing the protein of interest were pooled, concentrated using Vivaspin 20 Ultrafiltration Units (Sartorius, VS2011) and dialyzed against PBS (pH 7.4) at 4°C overnight using a SnakeSkin Dialysis Tubing (Thermo Fisher Scientific, 68100). The RBD was further polished by size exclusion chromatography (SEC) using a HiLoad 16/600 Superdex 200 pg column (Cytiva, 28-9893-35) equilibrated with PBS (pH 7.4). Both purified proteins were stored at −80°C until further use.

## Glycoproteomic analysis of Spike and RBD

Peptide mapping and glycoproteomic analysis of all samples were performed on in-solution proteolytic digests of the respective proteins by LC-ESI-MS(/MS). In brief, the pH of the samples was adjusted to pH 7.8 by the addition of 1 M HEPES, pH 7.8 to a final concentration of 100 mM HEPES, pH 7.8. The samples were then chemically reduced and S-alkylated using a final concentration of 10 mM dithiothreitol for 30 min at 56°C and a final concentration of 20 mM iodoacetamide for 30 min at room temperature in the dark, respectively. To maximize protein sequence-coverage of the analysis, proteins were digested with either Trypsin (Promega), a combination of Trypsin and GluC (Promega) or Chymotrypsin (Roche). Eventually, all proteolytic digests were acidified by addition of 10% formic acid to pH 2 and directly analysed by LC-ESI-MS(/MS) using an a capillary BioBasic C18 reversed-phase column (BioBasic-18, 150 × 0.32 mm, 5 μm, Thermo Scientific), installed in an Ultimate U3000 HPLC system (Dionex), developing a linear gradient from 95% eluent A (80 mM ammonium formate, pH 3.0, in HPLC-grade water) to 65% eluent B (80% acetonitrile in 80 mM ammonium formate, pH 3.0) over 50 min, followed by a linear gradient from 65 to 99% eluent B over 15 min, at a constant flow rate of 6 μl/min, coupled to a maXis 4G Q-TOF instrument (Bruker Daltonics; equipped with the standard ESI source). For (glyco)peptide detection and identification, the mass-spectrometer was operated in positive ion DDA mode (i.e. switching to MS/MS mode for eluting peaks), recording MS-scans in the *m/z*-range from 150 to 2,200 Th, with the 6 highest signals selected for MS/MS fragmentation. Instrument calibration was performed using a commercial ESI calibration mixture (Agilent). Site-specific profiling of protein glycosylation was performed using the dedicated Q-TOF data analysis software packages Data Analyst (Bruker Daltonics) and Protein Scape (Bruker Daltonics), in conjunction with the MS/MS search engine MASCOT (Matrix Sciences Ltd.) for automated peptide identification.

## ELISA assays to detect lectin binding to Spike and RBD

Briefly, 50 μl of full-length Spike-H6 (4 μg/ml, purified from HEK, diluted in PBS), RBD-H6 (2 μg/ml, purified from HEK, diluted in PBS) or human recombinant soluble ACE2 (hrsACE2; 4 μg/ml, purified from CHO, diluted in PBS, see (Monteil *et al*, 2020)) per well were used to coat a clear flat-bottom MaxiSorp 96-well plate (Thermo Fisher Scientific, 442404) for 2 h at 37°C. Thereafter, the coating solution was discarded and the plate was washed three times with 300 μl of wash buffer (1× TBS, 1 mM $CaCl_2$, 2 mM $MgCl_2$, 0.25% Triton X-100 (Sigma-Aldrich, T8787)). Unspecific

binding was blocked with 300 μl of blocking buffer (1× TBS, 1% BSA Fraction V (Applichem, A1391,0100), 1 mM CaCl₂, 2 mM MgCl₂ and 0.1% Tween-20 (Sigma-Aldrich, P1379)) for 30 min at 37°C. After removal of the blocking solution, 50 μl of either the mouse lectin-mIgG2a (10 μg/ml, diluted in blocking buffer), human CD209-hIgG1 (R&D Systems, 161-DC-050), human CD299-hIgG1 (R&D Systems, 162-D2-050), human CLEC4G-hIgG1 (Acro Biosystems, CLG-H5250-50ug) (10 μg/ml, diluted in blocking buffer), recombinant human ACE2-mIgG1 (2 μg/ml, diluted in blocking buffer, Sino Biological, 10108-H05H) or recombinant human ACE2-hIgG1 (2 μg/ml, diluted in blocking buffer, Sino Biological, 10108-H02H) were added for 1 h at room temperature. After washing for three times, 100 μl of 0.2 μg/ml HRP-conjugated goat anti-Mouse IgG (H + L) (Thermo Fisher Scientific, 31430) or goat anti-Human IgG (H + L) (Promega, W4031) antibodies were added for 30 min at room temperature. Subsequently, plates were washed as described above. To detect binding, 1 tablet of OPD substrate (Thermo Fisher Scientific, 34006) was dissolved in 9 ml of deionized water and 1 ml of 10× Pierce™ Stable Peroxide Substrate Buffer (Thermo Fisher, Scientific, 34062). 100 μl of OPD substrate solution was added per well and incubated for 15 min at room temperature. The reaction was stopped by adding 75 μl of 2.5 M sulfuric acid, and absorption was read at 490 nm. Absorption was measured for each lectin-Fc fusion protein tested against full-length Spike-H6, RBD-H6 or hrsACE2 and normalized against bovine serum albumin coated control wells. The binding of each lectin to Spike-H6, RBD-H6 or hrsACE2 was assessed in two independent ELISA experiments, each with 3 technical replicates.

### Protein denaturation and removal of N-glycans

To denature the full-length Spike-H6, 10 mM DTT was added to 40 μg/ml of protein. The samples were incubated at 85°C for 10 min. Thereafter, the denatured proteins were diluted to 4 μg/ml with PBS and a clear flat-bottom Maxisorp 96-well plate was coated with 50 μl per well for 2 h at 37°C. To remove the *N*-glycans from the full-length Spike-H6, 0.2 μg protein was denatured as above and adjusted to a final concentration of 1× Glycobuffer 2 containing 125 U PNGase F per μg (NEB, P0704S) in 50 μl. After incubation for 2 h at 37°C, the reaction was stopped by heat-inactivation for 10 min at 75°C. Spike proteins were then diluted to 4 μg/ml with PBS, and a clear flat-bottom Maxisorp 96-well plate was coated with 50 μl per well for 2 h at 37°C. ELISA protocols were performed as described above. To confirm the de-glycosylation of Spike proteins, 0.5 μg was loaded on an SDS–PAGE gel followed by a Coomassie staining.

### Surface plasmon resonance (SPR) measurements

A commercial SPR (BIAcore X, GE Healthcare, USA) was used to study the kinetics of binding and dissociation of lectin-Fc dimers to the trimeric full-length Spike in real time. Spike-H6 was immobilized on a Sensor Chip NTA (Cytiva, BR100034) via its His6-tag after washing the chip for at least 3 min with 350 mM EDTA and activation with a 1 min injection of 0.5 mM NiCl₂. 50 nM Spike were injected multiple times to generate a stable surface. For the determination of kinetic and equilibrium constants, the lectin samples (murine Clec4g, murine CD209c, human CLEC4G, human CD209) were injected at different concentrations (10 to 500 nM). As the binding of the lectins is Ca²⁺ dependent, lectins were removed from the surface by washing with degassed calcium free buffer (TBS, 0.1% Tween-20, pH = 7.4). The resonance angle was recorded at a 1 Hz sampling rate in both flow cells and expressed in resonance units (1 RU = 0.0001°). The resulting experimental binding curves were fitted to the "bivalent analyte model", assuming a two-step binding of the lectins to immobilized Spike. All evaluations were done using the BIAevaluation 3.2 software (BIAcore, GE Healthcare, USA).

### Single-molecule force spectroscopy (SMFS) measurements

For single-molecule force spectroscopy, a maleimide-Poly(ethylene glycol) (PEG) linker was attached to 3-aminopropyltriethoxysilane (APTES)-coated atomic force microscopy (AFM) cantilevers by incubating the cantilevers for 2 h in 500 μl of chloroform containing 1 mg of maleimide-PEG-N-hydroxysuccinimide (NHS) (Polypure, 21138-2790) and 30 μl of triethylamine. After 3 times washing with chloroform and drying with nitrogen gas, the cantilevers were immersed for 2 h in a mixture of 100 μl of 2 mM thiol-trisNTA, 2 μl of 100 mM EDTA (pH 7.5), 5 μl of 1 M HEPES (pH 7.5), 2 μl of 100 mM tris(carboxyethyl)phosphine (TCEP) hydrochloride and 2.5 μl of 1 M HEPES (pH 9.6) buffer and subsequently washed with HEPES-buffered saline (HBS). Thereafter, the cantilevers were incubated for 4 h in a mixture of 4 μl of 5 mM NiCl₂ and 100 μl of 0.2 μM His-tagged Spike trimers. After washing with HBS, the cantilevers were stored in HBS at 4°C (Oh *et al*, 2016). For the coupling of lectins to surfaces, a maleimide-PEG linker was attached to an APTES-coated silicon nitride surface. First, 2 μl of 100 mM TCEP, 2 μl of 1 M HEPES (pH 9.6), 5 μl of 1 M HEPES (pH 7.5) and 2 μl of 100 mM EDTA were added to 100 μl of 200 μg/ml Protein A-Cys (pro-1992-b, Prospec, NJ, USA) in PBS. The surfaces were incubated in this solution for 2 h and subsequently washed with PBS and 0.02 M sodium phosphate containing 0.02% sodium azide, pH = 7.0. Finally, 100 μl of 200 μg/ml lectin-Fc fusion proteins were added to the surfaces overnight.

Force–distance measurements were performed at room temperature (~25°C) with 0.01 N/m nominal spring constants (MSCT, Bruker) in TBS buffer containing 1 mM CaCl₂ and 0.1% Tween-20. Spring constants of AFM cantilevers were determined by measuring the thermally driven mean-square bending of the cantilever using the equipartition theorem in an ambient environment. The deflection sensitivity was calculated from the slope of the force–distance curves recorded on a bare silicon substrate. Determined spring constants ranged from 0.008 to 0.015 N/m. Force–distance curves were acquired by recording at least 1,000 curves with vertical sweep rates between 0.5 and 10 Hz at a z-range of typically 500–1,000 nm (resulting in loading rates from 10 to 10,000 pN/s), using a commercial AFM (5500, Agilent Technologies, USA). The relationship between experimentally measured unbinding forces and parameters from the interaction potential were described by the kinetic models of Bell (Bell, 1978) and Evans and Ritchie (Evans & Ritchie, 1997). In addition, multiple parallel bond formation was calculated by the Williams model (Williams, 2003) from the parameters derived from single-bond analysis. The binding probability was calculated from the number of force experiments displaying unbinding events over the total number of force experiments.

The probability density function (PDF) of unbinding force was constructed from unbinding events at the same pulling speed. For each unbinding force value, a Gaussian unitary area was computed with its centre representing the unbinding force and the width (standard deviation) reflecting its measuring uncertainty (square root of the variance of the noise in the force curve). All Gaussian areas from one experimental setting were accordingly summed up and normalized with its binding activity to yield the experimental PDF of unbinding force. PDFs are equivalents of continuous histograms as shown in Fig 3C.

## High-speed AFM (hsAFM) and data analysis

Purified SARS-CoV-2 trimeric Spike glycoproteins, murine Clec4g and CD209c and hCLEC4g and hCD209 were diluted to 20 µg/ml with imaging buffer (20 mM HEPES, 1 mM $CaCl_2$, pH 7.4), and 1.5 µl of the protein solution was applied onto freshly cleaved mica discs with diameters of 1.5 mm. After 3 min, the surface was rinsed with ˜15 µl imaging buffer (without drying) and the sample was mounted into the imaging chamber of the hsAFM (custom-built, RIBM, Japan). Movies were captured in imaging buffer containing 3 µg/ml of either Clec4g, CD209c, hCLEC4G or hCD209. An ultra-short cantilever (USC-F1.2-k0.15 nominal spring constant 0.15 N/m, Nanoworld, Switzerland) was used and areas of $100 \times 100$ nm containing single molecules were selected to capture the hsAFM movies at a scan rate of ˜150–300 ms per frame. During the acquisition of the movies, the amplitude was kept constant and set to 90–85% of the free amplitude (typically ˜3 nm). Data analysis was performed using the Gwyddion 2.55 software. Images were processed to remove background and transient noise. For volume measurements, a height threshold mask was applied over the protein structures with a minimum height of 0.25–0.35 nm to avoid excessive background noise in the masked area. The numbers of lectin molecules bound to the Spike trimers was calculated based on the measured mean volumes of the full-length Spike, the lectins, and the Spike-lectin complexes, averaged over the recorded time periods.

## AFM measured Spike binding to Vero E6 cells

Vero E6 cells were grown on culture dishes using DMEM containing 10% FBS, 500 units/ml penicillin and 100 µg/ml streptomycin, at 37°C with 5% $CO_2$. For AFM measurements, the cell density was adjusted to about 10-30% confluency. Before the measurements, the growth medium was exchanged to a physiological HEPES buffer containing 140 mM NaCl, 5 mM KCl, 1 mM $MgCl_2$, 1 mM $CaCl_2$ and 10 mM HEPES (pH 7.4). Lectins were added at the indicated concentrations. Using a full-length Spike trimer anchored to an AFM cantilever (described above), force–distance curves were recorded at room temperature on living cells with the assistance of a CCD camera for localization of the cantilever tip on selected cells. The sweep range was fixed at 3,000 nm, and the sweep rate was set at 1 Hz. For each cell, at least 100 force–distance cycles with 2000 data points per cycle and a typical force limit of about 30 pN were recorded.

## Glycan array analyses

The glycan microarrays were analysed by Asparia Glycomics (San Sebastian, Spain) and prepared as described previously (Brzezicka

*et al*, 2015). Briefly, 50 µM ligand solutions (1.25 nl, 5 drops, 250 pl drop volume) in sodium phosphate buffer (300 mM, 0.005% Tween-20, pH = 8.4) were spatially arrayed employing a robotic non-contact piezoelectric spotter (SciFLEXARRAYER S11, Scienion) onto N-hydroxysuccinimide (NHS) activated glass slides (Nexterion H, Schott AG). After printing, the slides were placed in a 75% humidity chamber for 18 h at 25°C. The remaining NHS groups were quenched with 50 mM solution of ethanolamine in sodium borate buffer (50 mM, pH = 9.0) for 1 h. The slides were washed with PBST (PBS/0.05% Tween-20), PBS and water, then dried in a slide spinner and stored at −20°C until use.

Glycan microarrays were compartmentalized using Proplate® 8 wells microarray gaskets, generating 7 independent subarrays per slide. Fusion proteins were diluted at a final concentration (10 µg/ml) in binding buffer (25 mM Tris, 150 mM NaCl, 4 mM $CaCl_2$ and 0.005% Tween-20 containing 0.5% bovine serum albumin). Lectins were applied to the microarrays and incubated at 4°C overnight with gentle shaking. The solutions were removed and arrays washed with binding buffer without BSA at room temperature. Interactions were visualized by the incubation of tetramethylrhodamine (TRITC) labelled secondary goat anti-mouse IgG antibodies (Fc specific; 1:1,000 dilution in binding buffer; Life Technologies) and goat anti-Human IgG (Fc specific)-Cy3 (1:1,000 dilution in binding buffer; Merck). Finally, slides were washed with binding buffer without BSA, dried in a slide spinner and scanned. Fluorescence was analysed using an Agilent G265BA microarray scanner (Agilent Technologies). The quantification of fluorescence was done using ProScanArray Express software (Perkin Elmer) employing an adaptive circle quantification method from 50 µm (minimum spot diameter) to 300 µm (maximum spot diameter). Average RFU (relative fluorescence unit) values with local background subtraction of four spots and standard deviation of the mean were recorded using Microsoft Excel and GraphPad Prism.

## Structural modelling

Structural models of the SARS-CoV-2 Spike protein were based on the model of the fully glycosylated Spike-hACE2 complex. Experimental structures deposited in the protein databank (PDB) were used to model the complex that is formed by the binding of SARS-CoV-2 Spike and ACE2 (Walls *et al*, 2020; Yan *et al*, 2020). RBD domain in complex with ACE2 was superimposed with Spike with one open RBD domain (PDB: 6VYB) and SWISS-MODEL was used to model missing residues in Spike (Waterhouse *et al*, 2018) (GenBank QHD43416.1). Glycan structures in agreement with the assignments of the current work were added using the methodology outlined by Turupcu and Oostenbrink (2017). The full model is available at the MolSSI / BioExcel COVID-19 Molecular Structure and Therapeutics Hub (https://covid.molssi.org//models/#spike-protein-in-complex-with-human-ace2-spike-spike-binding). For hCLEC4G, a homology model of residues 118–293 was constructed using Swiss-Model (Waterhouse *et al*, 2018) using residues 4–180 of chain A of the crystal structure of the carbohydrate recognition domain of DC-SIGNR (CD299) (PDB-code 1sl6, (Guo *et al*, 2004)). This fragment shows a sequence identity of 36% with hCLEC4G, and the resulting model showed an overall QMEAN value of −2.68. For mClec4g, the model consisted of residues 118–294, with a sequence identity of 38% to the same template model. The resulting QMEAN value was −2.88. A

calcium ion and the bound Lewis × oligosaccharide of the template were taken over into the model, indicating the location of the carbohydrate-binding site. For hCD209, the crystal structure of the carbohydrate recognition domain of CD209 (DC-SIGN) complexed with Man4 (PDB code 1sl4, (Guo *et al*, 2004)) was used. To identify binding sites of hCLEC4G and hCD209 to the Spike-hACE2 complex, a superposition of the bound carbohydrates with the glycans on Spike was performed. For hCLEC4G, we used the complex glycans at N343 of the third monomer of Spike, with the receptor binding domain in an "up" position, while N343 glycans on monomer 1 and 2 were modelled with the receptor binding domain in a "down" position. For hCD209, we used the high-mannose glycan at position N234 in monomer 1–3 of Spike, respectively. These glycan structures were chosen in accordance with the full-length Spike glycoproteome.

## SARS-CoV-2 infections

All experiments with live SARS-CoV-2 were performed in a BSL-3 laboratory, following all appropriate safety and security protocols approved at the Public Health Agency of Sweden. Vero E6 cells ($5 \times 10^4$ cells per well) or Calu3 cells ($1 \times 10^5$ cells per well, both from ATCC, Manassas, VA) were seeded in 48-well plates (Sarstedt, 83.3923) in DMEM containing 10% FBS. 24 h post-seeding, different concentrations of lectins were mixed with $10^3$ PFU of virus (1:1) to a final volume of 100 µl per well in DMEM (resulting in a final concentration of 5% FBS). After incubation for 30 min at 37°C, Vero E6 or Calu3 cells were infected either with mixes containing lectins/SARS-CoV-2 or SARS-CoV-2 alone (indicated as mock-treated in the Figures) or DMEM with 5% FBS (no virus control). 15 h post-infection, supernatants were removed, cells were washed three times with PBS and then lysed using TRIzol Reagent (Thermo Fisher Scientific, 15596026). The qRT–PCR for the detection of viral RNA was performed as previously described (Monteil *et al*, 2020). Briefly, RNA was extracted using the Direct-zol RNA MiniPrep kit (Zymo Research, R2051). The qRT–PCR was performed for the SARS-CoV-2 E gene and RNase P was used as an endogenous gene control to normalize viral RNA levels to the cell number. Lectins were independently tested for cellular toxicity in an ATP-dependent assay (Cell-Titer Glo, Promega) and cells found to exhibit > 80% viability up to a concentration of 200 µg/ml. Cell lines have been authenticated by STR profiling and were tested negative for mycoplasma.

The following PCR Primers were used:
SARS-CoV2 E-gene:

Forward primer: 5′-ACAGGTACGTTAATAGTTAATAGCGT-3′
Reverse primer: 5′-ATATTGCAGCAGTACGCACACA-3′
Probe: FAM-ACACTAGCCATCCTTACTGCGCTTCG-QSY
    RNase P:
Forward primer: 5′-AGATTTGGACCTGCGAGCG-3′
Reverse primer 5′-GAGCGGCTGTCTCCACAAGT-3′
Probe: FAM-TTCTGACCTGAAGGCTCTGCGCG-MGB

## Competition ELISA

MaxiSorp 96-well plates (Thermo Fisher Scientific, 442404) were coated with 50 µl of full-length trimeric Spike-H6 (4, 1, 0.5 or 0.1 µg/ml, purified from HEK, diluted in PBS) for 2 h at 37°C. The

coating solution was discarded and the plate washed three times with 300 µl of washing buffer (1× TBS, 1 mM CaCl$_2$, 2 mM MgCl$_2$, 0.02% Tween-20 (Sigma-Aldrich, P1379)). Unspecific binding was blocked with 300 µl of blocking buffer (1× TBS, 1% BSA Fraction V (Applichem, A1391,0100), 1 mM CaCl$_2$, 2 mM MgCl$_2$ and 0.1% Tween-20 (Sigma-Aldrich, P1379)) for 30 min at 37°C. For pilot studies, 100 µl hrsACE2 (0, 0.002, 0.01, 0.05, 0.1, 0.5, 2.5, 5 or 25 nM, purified from CHO, diluted in blocking buffer, see (Monteil *et al*, 2020)) were applied. For the competition ELISA, 100 µl of hrsACE2 (at a final concentration of 0.1 nM) and lectin-Fc (at a final concentration of 0, 0.05, 0.1, 0.5, 1 or 2.5 µM) in blocking buffer were co-incubated for 1 h at room temperature. After three washes, detection antibodies and Streptavidin-HRP of the Human ACE2 DUOSet ELISA Kit (R&D Systems, DY933-05) were used according to the manufacturer's protocol. To detect binding, 1 tablet of OPD substrate (Thermo Fisher Scientific, 34006) was dissolved in 9 ml of deionized water and 1 ml of 10× Pierce™ Stable Peroxide Substrate Buffer (Thermo Fisher, Scientific, 34062). 100 µl of OPD substrate solution was added per well and incubated for 1 h at room temperature. The reaction was stopped by adding 75 µl of 2.5 M sulfuric acid and absorption was read at 490 nm. Absorptions were normalized against 1% bovine serum albumin coated control wells.

## Recombinant expression of SARS-CoV-2 Spike N343Q

A pTT28 mammalian expression vector (National Research Council, NRC, Ottawa, Canada) encoding the soluble spike glycomutant N343Q was generated by inverse PCR using phosphorylated primers and the pTT28-S wt plasmid template. Following site-directed mutagenesis PCR, the plasmid template was digested with *Dpn*I, the amplified mutant vector was gel-purified, self-ligated and transformed into competent cells Neb5α E.coli cells (both New England Biolabs, Ipswich, MA). Soluble S-wt and S-N343Q were expressed in 125-ml shake flasks in triplicates using HEK293-6E cells (licensed from National Research Council, NRC, Ottawa, Canada) according to established protocols (Klausberger *et al*, 2021). Five days post-transfection cells were pelleted by low-speed centrifugation (500 *g*, 10 min), pellets were washed once with ice-cold PBS, and supernatants and pellets were used for protein and RNA extraction.

Protein of supernatants were precipitated using trichloroacetic acid (TCA). Briefly, 1 ml of expression supernatants was mixed with 250 µl TCA and incubated on ice for 30 min. Proteins were pelleted by centrifugation at 16,000 *g* for 10 min, and pellets were washed with 0.5 ml ice-cold ethanol followed by 0.5 ml ice-cold acetone. Pellets were allowed to air-dry and were directly dissolved in SDS sample buffer.

Proteins from cell pellets were extracted by ml Pierce™ RIPA buffer (#89900, Thermo Fisher Scientific, Waltham, MA) supplemented with SIGMAFAST protease inhibitor cocktail (#8830, Sigma-Aldrich, St. Louis, MI) as per the manufacturers' protocols. Recombinant spike proteins were detected via Western blot using a biotinylated His-Tag-specific mAb (#MA1-21315-BTIN; Invitrogen, Carlsbad, CA) followed by HRP-conjugated Streptavidin (#N100; Thermo Fisher Scientific, Waltham, MA).

For RNA extraction, cell pellets were incubated with RLT Plus Buffer containing 1% 2-Mercaptoethanol followed by RNA extraction

using the Qiagen RNA easy plus kit (Qiagen, Hilden, GER). Eluted RNA was treated with Turbo DNase (Invitrogen, Carlsbad, CA) to remove genomic DNA and Plasmids. RNA was quantified using NanoDrop 260nm absorbance measurements. Poly-A cDNA was generated using Oligo(dT) primers and GoScript Reverse transcriptase (Promega). Additionally, control reactions were performed in the absence of reverse transcriptase. Detection of SARS-CoV-2 spike or internal human reference control gene (EIF2B2) cDNA was done using the Luna qPCR reaction mixture (NEB). For the internal control, 1 μl of a 20× IDT Primer Probe mix was added. The reactions were amplified in a CFX96 real-time qPCR instrument (Bio-Rad).

| SARS-CoV-2 Spike-F1 | 5′-CAGCTACCAGACACAGACAAA-3′ |
| SARS-CoV-2 Spike-R1 | 5′-GCGATAGAGTTGTTGGAGTAGG-3′ |
| SARS-CoV-2 Spike-F2 | 5′-GATCGCCGACTACAACTACAA-3′ |
| SARS-CoV-2 Spike-R2 | 5′-GAACAGCCGGTACAGGTAAT-3′ |

EIF2B2 Primer Probe assay (IDT, Hs.PT.58.39825985, 5′ HEX/ZEN/3′ IBFQ).

## Data availability

This study includes no data deposited in external repositories.

**Expanded View** for this article is available online.

## Acknowledgements

We thank all members of the Penninger laboratory for helpful discussions and technical support. Moreover, we thank all members of the Molecular Biology Service and the VBCF Protein Technologies Facility. We thank Florian Krammer (Icahn School of Medicine at Mount Sinai, NY, United States) for providing the constructs used for production of recombinant Spike and RBD. Purified Spike and RBD as well as transfection-grade pCAGGS plasmids were obtained from the reagent repository of the BOKU COVID-19 Initiative. The authors thank Daniel Maresch (BOKU Core Facility Mass Spectrometry) for assistance with glycan analysis. Figures 1A and 3A, Appendix Fig S1A created with BioRender.com. J.M.P. and the research leading to these results has received funding from the T. von Zastrow foundation, the FWF Wittgenstein award (Z 271-B19), the Austrian Academy of Sciences, the Innovative Medicines Initiative 2 Joint Undertaking (JU) under grant agreement No 101005026, and the Canada 150 Research Chairs Program F18-01336 as well as the Canadian Institutes of Health Research COVID-19 grants F20-02343 and F20-02015. D.H. is supported by the T. von Zastrow foundation, S.M. is funded by the European Union's Horizon 2020 research and innovation programme under the Marie Sklodowska-Curie grant agreement No 841319. We further acknowledge financial support from the Austrian National Foundation for Research, Technology, and Development and Research Department of the State of Upper Austria (Y.J.O), from the FWF projects V584 (Y.J.O), P31599 (R.Z.), I3173 (P.H., L.H.), the WWTF grant LS19-029 (P.H, D.C.) and grant COV20-015 (C.O.), the ÖAW fellowship STIP13202002 (L.H.), and the European Union's Horizon research and innovation programme (H2020-MSCA-ITN-2016) under the Marie Sklodowska-Curie grant agreement No. 721874 (P.H., D.C.). A.M. and V.M. have received funding from the Innovative Medicines Initiative 2 JU under grant agreement no. 101005026. JU receives support from the European Union's Horizon 2020 research and innovation programme and EFPIA.

## Author contributions

DH, SM and JMP conceived and designed the study; YJO and PH conceived and coordinated the SPR and AFM studies; YJO performed the lectin force spectroscopy measurements and data analysis; RZ performed the binding activity measurements and data analysis; DC performed the high-speed AFM measurements and data analysis; LH performed the SPR measurements and data analysis; VM, EE and AM designed, performed and analysed the SARS-CoV-2 infection experiments; EL, MK, MJK and LM set-up and performed the purification of the Spike and RBD and verified expression quality; CG-G, FA and JS designed, performed and analysed the glycosylation of the Spike and RBD; DH, GW, MN, AC, GJ and MT designed, set-up and purified the lectin-Fc proteins; DH, AH and SM performed all ELISA experiments; CO performed the modelling; YJO, RZ, DC and LH wrote the original SPR and AFM part of this manuscript with guidance and edits from PH; DH, SM and JMP wrote the manuscript. All authors read and reviewed of the manuscript.

## Conflict of interest

A patent is being prepared to use CLEC4G as potential therapy for COVID-19. J.M.P. is shareholder and board member of Apeiron Biologics that is developing soluble ACE2 for COVID-19 therapy.

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
