## [Review Process File · The EMBO Journal]

Identification of lectin receptors for conserved SARS-CoV-2 glycosylation sites

David Hoffmann, Stefan Mereiter, Yoo Jin Oh, Vanessa Monteil, Elizabeth Elder, Rong Zhu, Daniel Canena, Lisa Hain, Elisabeth Laurent, Clemens Gruenwald-Gruber, Miriam Klausberger, Gustav Jonsson, Max Kellner, Maria Novatchkova, Melita Ticevic, Antoine Chabloz, Gerald Wirnsberger, Astrid Hagelkruys, Friedrich Altmann, Lukas Mach, Johannes Stadlmann, Chris Oostenbrink, Ali Mirazimi, Peter Hinterdorfer, and Josef Penninger

DOI: [10.15252/embj.2021108375](https://doi.org/10.15252/embj.2021108375)

Corresponding author: Josef Penninger (josef.penninger@imba.oeaw.ac.at)

Review Timeline:

Submission Date:	31st Mar 21
Editorial Decision:	29th Apr 21
Revision Received:	23rd Jun 21
Editorial Decision:	9th Jul 21
Revision Received:	19th Jul 21
Accepted:	28th Jul 21

Editor: Karin Dumstrei

Transaction Report:

Dear Josef,

Thank you for submitting your manuscript to The EMBO Journal. Your study has now been seen by three referees and their comments are provided below.

As you can see from the comments below, the referees appreciate the analysis and find the study important. I would therefore like to invite a revised version. The referees raise a number of relevant points that should be addressed. I think it would be helpful to discuss the raised points further and I am available to do so via email or video.

Thank you for the opportunity to consider your work for publication. I am looking forward to discussing the revisions further

with best wishes

Karin

Karin Dumstrei, PhD
Senior Editor
The EMBO Journal

When assembling figures, please refer to our figure preparation guideline in order to ensure proper formatting and readability in print as well as on screen:
<https://bit.ly/EMBOPressFigurePreparationGuideline>

- a point-by-point response to the referees' comments, with a detailed description of the changes made (as a word file).
- a word file of the manuscript text.
- individual production quality figure files (one file per figure)
- a complete author checklist, which you can download from our author guidelines (<https://www.embopress.org/page/journal/14602075/authorguide>).

- Expanded View files (replacing Supplementary Information)

Further information is available in our Guide For Authors:

The revision must be submitted online within 90 days; please click on the link below to submit the revision online before 28th Jul 2021.

Link Not Available

Referee #1:

- general summary

The author found and proved the lectin receptors for SARS-CoV-2 glycosylation sites. CLEC4g and CD209c may become two potential target of broad-spectrum coronavirus entry inhibitor for the development of antiviral drugs. It's a very meaningful basic research to deal with a pandemic in the long run.

- minor concerns that should be addressed

1. In this study, 143 CRDs were expressed and purified, a huge workload. the screening results could be impact by the quality of proteins, especially the conformation of protein. How to control before detection? In table S2, there was little variation in ELISA, n = 2. Is this a repeat of 2 wells or two independent tests?

2. To prove the interaction with glycosylation sites, enzymolysis was used. The specificity of enzymatic hydrolysis is an influencing factor. How to control it? If we use the pseudovirus research of mutant strain, we can further identify the action site

3. Reg1, mgl2 or asgr1, one or more of them should be selected for the follow-up study, as the control of the follow-up validation test, and the specificity and accuracy of the CLEC4g and CD209c proteins should be fully discussed.

- suggestions for improving the study

4. The results of AFM are shown in Figure 3, but the results are not clearly displayed, can the author design a more vivid model diagram?

Referee #2:

In their manuscript "Identification of lectin receptors for conserved 1 SARS-CoV-2 glycosylation sites " Hofmann and colleagues provide a detailed analysis of murine and human lectin binding to SARS-CoV2 Spike protein that leads to the identification of potential antiviral lectins. The manuscript is clearly written and the data is well-presented and discussed. There are, however, several major critique points that need to be addressed.

1. Why did the authors use murine instead of human lectins to start with the investigation? It would have made more sense to use the human lectins from the start. Please explain the rationale.
2. All experiments are performed with Spike proteins produced in 293T-derived cells. It is known that 293Ts produce aberrant glycosylation patterns (less sialylation, more fucosylation, more tri- and tetra-antennar glycosylation patterns (e.g. Böhm et al, BMC biotechnology)). HIV gp120 glycosylation pattern is markedly different when expressed in 293T-derived cells or in PBMCs (e.g. Pritchard et al, JV, 2015). The authors need to provide evidence that the Spike protein glycopattern they observe is the authentic glycopattern found in virions. This is even more relevant as single protein expression of Spike and transport through the Golgi might lead to different glycosylation pattern than after virion assembly and budding into the ERGIC.
3. Similarly, the infection studies are performed in Vero cells only. Validation in primary human airway epithelia is needed as also in Vero cells the glycosylation pattern might differ from the in vivo situation in humans.
4. To prove specificity of the identified lectins, the authors should use the pseudotyped viruses containing the point mutations in the glycosylation sites, or even better, recombinant SARS-CoV2 containing the point mutations.
5. Does SARS-CoV2 acquire escape/rescue mutations when cultures and passaged in the presence of low amounts of the antiviral lectins?
6. The authors state in the main manuscript that they use "full-length trimeric spike" but fail to point out that in their construct the furin cleavage site is mutated and it is the stabilized pre-fusion spike containing 2 proline mutations. This is an important information that has to be mentioned in the main text. The authors should also indicate if the mutation of the furin cleavage site likely affects the overall structure and possibly the glycosylation or not.
7. Line 57: Animal cells do not have a cell wall.

Referee #3:

Hoffmann et al. screen a library of 143 recombinant murine lectin-Fc proteins for RBD and Spike binding by ELISA. From this screen, two lectins are selected for further binding analysis by SPR and AFM and for the ability to inhibit SARS-CoV-2 infection of Vero E6 cells. The identification of two lectins that can bind glycans found on Spike is an interesting finding but is it significant? There are no experiments testing whether these lectins are important for SARS-CoV-2 replication, whilst the micromolar affinity of binding means that they are not potent enough to serve as therapeutic inhibitors.

Specific comments:

1. The affinities of the two lectins are μM as measured by SPR but binding inhibition was observed by AFM at 20-70 nM - why the difference?
2. ELISA has been used to measure both Spike:Lectin binding and Spike:ACE2 binding. It would be straightforward to repeat this assay and perform competition experiments to determine IC50s for lectin inhibition of ACE2 binding. This would provide important confirmation that lectins can compete for ACE2 binding.
3. Previous studies have implicated DC-SIGN as a physiologically important lectin for SARS-CoV-2 infection. In order to show that the new lectins identified here are physiologically relevant, similar experiments should be performed. For instance, by depleting the lectins using siRNA and determining whether this decreases SARS-CoV-2 infection.
4. Why was the original lectin library made with murine proteins? Could the screen have failed to pick up positives because only human equivalents bind? This is strongly relevant as SARS-CoV-2 does not infect mice and a transgenic with human ACE2 is required.
5. Related to the above, Figure 4 is simply a repeat of 2&3 but with human lectins.
6. SDS PAGE of expressed RBD and full-length Spike should be shown.
7. Figure 6C&D address the most important question in the manuscript, namely can the identified lectins inhibit SARS-CoV-2 infection? This needs to be expanded into its own figure and a more thorough and convincing dataset collected. Specifically:

A/ Biological replicates are required for 6D

B/ The individual data points in 6C & 6D should be plotted to allow a better assessment of variability.

C/ The data should be expressed as genome copies normalised to a host gene control and not as fold-change.

D/ What are the controls? What is mock-treated? The methods refer to mock-infected, which suggests no virus was added in this condition, but not a mock-treated condition.

E/ The experiments should be done with a no virus control and also a control using a non-binding lectin (from the panel of 143 expressed lectins).

Point by point reply to Referee #1: Please note, all changes are marked in the revised manuscript.

The author found and proved the lectin receptors for SARS-CoV-2 glycosylation sites. CLEC4g and CD209c may become two potential target of broad-spectrum coronavirus entry inhibitor for the development of antiviral drugs. It's a very meaningful basic research to deal with a pandemic in the long run.

We thank the reviewer for the kind remarks.

Minor concerns that should be addressed.

1. In this study, 143 CRDs were expressed and purified, a huge workload. The screening results could be impacted by the quality of proteins, especially the conformation of protein. How to control before detection? In table S2, there was little variation in ELISA, n = 2. Is this a repeat of 2 wells or two independent tests?

Each lectin was assessed in 2 independent ELISAs, and within each ELISA the binding was quantified in triplicates (3 wells). We thank the reviewer for this comment and further clarified the replicate number in the material and methods section (lines 521 to 523) and in the legend of table S3.

2. To prove the interaction with glycosylation sites, enzymolysis was used. The specificity of enzymatic hydrolysis is an influencing factor. How to control it? If we use the pseudovirus research of mutant strain, we can further identify the action site.

We fully agree with the reviewer that the application of N343 mutant Spike would be an elegant complementation of our data. Unfortunately, mutation of N343 leads to an insoluble/instable protein product. We have now included this experimental data as a new supplementary figure 9 in the revised manuscript. We believe that this is an important finding as it explains observations made by others, such as the complete loss of ACE2 binding when this particular Spike protein site is mutated. Please see also our discussion (lines 298 to 302).

3. Reg1, mgl2 or asgr1, one or more of them should be selected for the follow-up study, as the control of the follow-up validation test, and the specificity and accuracy of the CLEC4g and CD209c proteins should be fully discussed.

We thank the reviewer for this suggestion. We have now included Asgr1 in the infection studies (please see revised Figure 6C, D). Note, Asgr1 was the lectin that strongly bound to RBD but not to full-length trimer Spike in our lectin screen. We were therefore able to further underline the validity of our screen in identifying SARS-CoV-2 Spike binding lectins, also highlighting the importance to use trimeric full-length Spike rather than RBD for screening or drug studies.

4. The results of AFM are shown in Figure 3, but the results are not clearly displayed, can the author design a more vivid model diagram?

We thank the reviewer for the feedback. We have added an improved model diagram to supplementary Figure 3 and 4, which should facilitate the interpretation of our results.

Point by point reply to Referee #2: Please note, all changes are marked in the revised manuscript.

In their manuscript "*Identification of lectin receptors for conserved 1 SARS-CoV-2 glycosylation sites*" Hofmann and colleagues provide a detailed analysis of murine and human lectin binding to SARS-CoV2 Spike protein that leads to the identification of potential antiviral lectins. The manuscript is clearly written and the data is well-presented and discussed.

We thank the reviewer for the positive feedback.

There are, however, several major critique points that need to be addressed.

1. Why did the authors use murine instead of human lectins to start with the investigation? It would have made more sense to use the human lectins from the start. Please explain the rationale.

The reason was pragmatic. We had the murine lectin library in place when SARS-CoV-2 emerged. We therefore proceeded with the murine lectin library and confirmed hits by applying human homologues. We are currently cloning a human library but to finish this will take another 5-6 months, including purifying sufficient amounts of lectins for screens.

2. All experiments are performed with Spike proteins produced in 293T-derived cells. It is known that 293Ts produce aberrant glycosylation patterns (less sialylation, more fucosylation, more tri- and tetra-antennar glycosylation patterns (e.g. Böhm et al, BMC biotechnology)). HIV gp120 glycosylation pattern is markedly different when expressed in 293T-derived cells or in PBMCs (e.g. Pritchard et al, JV, 2015). The authors need to provide evidence that the Spike protein glycopattern they observe is the authentic glycopattern found in virions. This is even more relevant as single protein expression of Spike and transport through the Golgi might lead to different glycosylation pattern than after virion assembly and budding into the ERGIC.

We thank the reviewer for this important comment. Indeed, the authenticity of Spike glycosylation is an important factor. Our data shows that lectins in accordance with the screen were able to inhibit life SARS-CoV-2 virus on VeroE6 and Calu3 cells, supporting the relevance of our data. In addition, in a recent manuscript Brun and colleagues (doi/10.1021/acscentsci.1c00058) have shown that the glycosylation of SARS-CoV-2 Spike derived from human lung cells reflects the glycosylation described in our manuscript. We have further addressed this point in the discussion of our revised paper (lines 284 to 288).

3. Similarly, the infection studies are performed in Vero cells only. Validation in primary human airway epithelia is needed as also in Vero cells the glycosylation pattern might differ from the in vivo situation in humans.

We fully agree with the referee and have included this important experiment in the revised Figure 6 of the manuscript. The new data confirm and strengthen the results in our first submission.

4. To prove specificity of the identified lectins, the authors should use the pseudotyped viruses containing the point mutations in the glycosylation sites, or even better, recombinant SARS-CoV2 containing the point mutations.

We fully agree with the reviewer that the application of N343 mutant Spike would be an elegant complementation of our data. Unfortunately, mutation of N343 leads to an insoluble/instable protein product. We have now included this experimental data in the revised manuscript. We believe that this is an important finding as it explains observations made by others, such as the complete loss of ACE2 binding when this particular Spike protein site is mutated.

5. Does SARS-CoV2 acquire escape/rescue mutations when cultures and passaged in the presence of low amounts of the antiviral lectins?

This is an interesting question. However, this would require substantially higher amounts of lectins than we currently can provide. We therefore believe that this question is out of scope of the current manuscript and hope that the referee agrees that we address this issue in future experiments.

6. The authors state in the main manuscript that they use "full-length trimeric spike" but fail to point out that in their construct the furin cleavage site is mutated and it is the stabilized pre-fusion spike containing 2 proline mutations. This is an important information that has to be mentioned in the main text. The authors should also indicate if the mutation of the furin cleavage site likely affects the overall structure and possibly the glycosylation or not.

We thank the reviewer for bringing this to our attention. We have addressed these important comments in the revised paper and have adapted the manuscript in accordance to the reviewer's comment (lines 99 to 100 and 284 to 288).

7. Line 57: Animal cells do not have a cell wall.

We thank the reviewer for bringing this error to our attention, which has been corrected.

Point by point reply to Referee #3: Please note, all changes are marked in the revised manuscript.

Hoffmann et al. screen a library of 143 recombinant murine lectin-Fc proteins for RBD and Spike binding by ELISA. From this screen, two lectins are selected for further binding analysis by SPR and AFM and for the ability to inhibit SARS-CoV-2 infection of Vero E6 cells. The identification of two lectins that can bind glycans found on Spike is an interesting finding but is it significant? There are no experiments testing whether these lectins are important for SARS-CoV-2 replication, whilst the micromolar affinity of binding means that they are not potent enough to serve as therapeutic inhibitors.

Specific comments:

1. The affinities of the two lectins are μM as measured by SPR but binding inhibition was observed by AFM at 20-70 nM - why the difference?

We thank the reviewer for this comment. The discrepancy of the two values stems from monomeric vs dimeric binding. Specifically, we used in our study dimeric lectin-Fc fusion proteins. The SPR experimental binding curves showed very good fittings to the “bivalent analyte model” (Traxler et al., 2017), which assumes two-step binding of the lectin dimers to adjacent immobilized Spike trimer binding sites and can be used to resolve the two binding steps individually. The kinetic association ($k_{a,1}$), kinetic dissociation ($k_{d,1}$), and equilibrium dissociation ($K_{d,1}$) binding constants of the first monomeric binding step shown in Table 1, therefore reflect the values of a single lectin bond. In contrast, the binding constants derived from the inhibition experiments are binding inhibition values for bi-valent binding of the lectin dimers, in which the two binding steps are “summed up”. Thus, in line with our expectation, the K_i is significantly lower than $K_{d,1}$. We have now clarified this important point in the revised manuscript (Lines 295 to 298).

2. ELISA has been used to measure both Spike:Lectin binding and Spike:ACE2 binding. It would be straightforward to repeat this assay and perform competition experiments to determine IC50s for lectin inhibition of ACE2 binding. This would provide important confirmation that lectins can compete for ACE2 binding.

We thank the reviewer for this suggestion. We have now performed these competition experiments. The new results are shown in Supplementary Figure 6 of the revised manuscript, confirming the conclusions of our manuscript.

3. Previous studies have implicated DC-SIGN as a physiologically important lectin for SARS-CoV-2 infection. In order to show that the new lectins identified here are physiologically relevant, similar experiments should be performed. For instance, by depleting the lectins using siRNA and determining whether this decreases SARS-CoV-2 infection.

We thank the reviewer for this comment. The murine lectins clec4g and cd209c as well as human CLEC4G are expressed by a very specific endothelial cells, namely lymph endothelium and liver sinusoidal endothelium. To our knowledge no cell line is yet available that expresses these lectins and therefore knockout/knockdown experiments cannot be performed at the moment. We are actually trying to generate lymphoid organoids to address such a function, but this, we trust, can be done for a follow-up story.

4. Why was the original lectin library made with murine proteins? Could the screen have failed to pick up positives because only human equivalents bind? This is strongly relevant as SARS-CoV-2 does not infect mice and a transgenic with human ACE2 is required.

We had the murine lectin library in place when SARS-CoV-2 emerged. We therefore proceeded with the murine lectin library and confirmed hits by applying human homologues. We of course realize that an inherent limitation of all screens is that false negatives cannot be excluded. We are currently cloning a human library but to finish this will take another 5-6 months, including purifying sufficient amounts of lectins for screens.

5. Related to the above, Figure 4 is simply a repeat of 2&3 but with human lectins.

We performed the screen using mouse lectins. It was therefore essential to bring these data to the human arena and provide human relevance. This is critical data as we confirmed with a broad range of methods that the human homologues behave similar to the murine counterparts from our screen.

6. SDS PAGE of expressed RBD and full-length Spike should be shown.

In accordance with the comment, we have updated supplementary figure 2 showing now the SDS-PAGE image of RBD and full-length Spike, with and without deglycosylation.

7. Figure 6C&D address the most important question in the manuscript, namely can the identified lectins inhibit SARS-CoV-2 infection? This needs to be expanded into its own figure and a more thorough and convincing dataset collected. Specifically:

We fully agree with the reviewer. We have included a large set of additional infection experiments, with additional replicates, non-binding lectins and, importantly, also using a human lung airway cell line.

A/ Biological replicates are required for 6D

In accordance with the reviewer's comment, we have included additional replicates and included the new data in Figure 6C.

B/ The individual data points in 6C & 6D should be plotted to allow a better assessment of variability.

We thank the reviewer for this suggestion. We have now plotted the individual data points in Figure 6C, and the new Figure 6D.

C/ The data should be expressed as genome copies normalised to a host gene control and not as fold-change.

We thank the reviewer for this comment. We agree with the reviewer that such normalization of viral RNA copies is important; we have normalized our data to the RNase P host gene control, following the request of the reviewer. This has also been highlighted in material and methods. We showed our data as fold change in our previous publications Monteil V. et al, Cell, 2020; PMID: 32333836 and Monteil V. et al, EMBO Mol Med, 2021; PMID: 33179852. We believe this is also a very good approach to demonstrate the relation the viral RNA changes in treated and mock treated cells.

D/ What are the controls? What is mock-treated? The methods refer to mock-infected, which suggests no virus was added in this condition, but not a mock-treated condition.

The controls are mock-treated, meaning they were infected with SARS-CoV2 but without the addition of lectins. The method was modified to make sure that this is absolutely clear (page 29, Line 708).

E/ The experiments should be done with a no virus control and also a control using a non-binding lectin (from the panel of 143 expressed lectins).

We thank the reviewer for this comment. No virus controls are included in our experiments. We have further clarified the methodology in the material and methods section (line 708) as well as in the figure legend of Figure 6. Following the request of the reviewer, we have now added a control lectin, Asgr1, that has been shown in our screen to be a non-binder to Spike. Interestingly, Asgr1 actually bound to the isolated RBD domain but not to trimeric Spike, validating our approach and highlighting that it is critical to use full length trimeric Spike and not RBD for such screens.

Dear Josef,

Thank you for submitting your revised version to The EMBO Journal. Your study has now been seen by referees # 2 and 3 and both are happy with the introduced changes. Referee #2 has a remaining question regarding the way the P values were calculated. Can you please clarify this? This was also picked up by our data editor - please see last point in list below.

Besides this just a few editorial points that needs to be sorted out.

We are missing 3-5 keywords

We are missing a Data Availability section. This is the place to enter accession numbers etc. If no data is generated that needs to be deposited in a database- please see our guide to authors then please state: This study includes no data deposited in external repositories. Please place it after the Materials and methods and before Acknowledgements

Also, can you add a statement if you have approval for working with SARS-CoV-2 virus in your lab and if laboratory security protocols was applied. I think good to include it for full transparency.

We don't allow data not shown (pgd 29 + 44). Can you expand on this in the text?

Figure callouts are missing to Fig S7 and Fig S9 panels. Please double check.

Movies: The nomenclature needs correcting to 'Movie EV#' - please also correct callouts in text. They need to be ZIPed with their legends. The legends should be removed from the Manuscript file.

You have 9 supplemental figure files. You can either turn them into EV figures (but we can only have 5) or make them into an appendix. You can also do a combination of both. Please see guide to authors. The appendix needs a ToC, appendix figure legends should be removed from the Manuscript file. Please also include the 4 supplementary tables into the Appendix file and remove their legends from MS file. See guide to authors for correct nomenclature for EV and appendix figures.

The snapshots shown in S3E and F are based upon the same movies in Figure 3E and F - correct? You refer to Figure 3C in the Fig. S3. Please double check.

Our publisher has done their pre-acceptance checks on the manuscript. Please see the comments on figure legends. The manuscript file is called Data edited MS file. Take a look at the word file. Please in particular check: N=2 in Fig 3D, S4A and S8B/C. N = 1 independent experiment in Fig 6C. The number of replicates appears to be missing from Fig EV3D, S4E, S6 and S8A. For N =2 better to display both data points. For the N=1 in Fig 6C as also mentioned by referee #2 - lets discuss.

Let me know if you have any further questions.

With best wishes

Karin

Karin Dumstrei, PhD
Senior Editor
The EMBO Journal

Guide For Authors: <https://www.embopress.org/page/journal/14602075/authorguide>

The revision must be submitted online within 90 days; please click on the link below to submit the revision online before 7th Oct 2021.

Link Not Available

Referee #2:

The authors addressed most of my concerns.

However, the validation has been performed in a cell line and not primary human airway epithelia as suggested.

It would have been sufficient to do the experiment only once in triplicate in primary cells, but as the infection experiment was done in a cell line, $n=1$ in triplicate and calculating p values from triplicates is not sufficient. In general, p values should be calculated from independent experiments to account for biological variation.

Referee #3:

The authors have thoroughly addressed my comments and provided additional data. I am happy to recommend publication and congratulate the authors on their study.

Reply to Referee #2

The authors addressed most of my concerns. However, the validation has been performed in a cell line and not primary human airway epithelia as suggested. It would have been sufficient to do the experiment only once in triplicate in primary cells, but as the infection experiment was done in a cell line, n=1 in triplicate and calculating p values from triplicates is not sufficient. In general, p values should be calculated from independent experiments to account for biological variation.

We thank the reviewer and fully agree that biological replicates are important to account for biological variation. Importantly, the experiments in Fig 6 were repeated in multiple replicates (both technical and biological) using two different cell lines, strengthening and validating our conclusions. To enhance the reader's awareness on the nature of the replicates and to avoid any confusion, we have added the detailed information in the figure legend.

Reply to Referee #3

The authors have thoroughly addressed my comments and provided additional data. I am happy to recommend publication and congratulate the authors on their study.

We thank the referee for the positive comment.

Dear Josef,

Thanks for submitting your revised manuscript to The EMBO Journal. I have now had a chance to take a look at everything and all looks good.

I am therefore very pleased to accept the manuscript for publication here.

Congratulations on a nice study.

with best wishes

Karin

Karin Dumstrei, PhD
Senior Editor
The EMBO Journal

Please note that it is EMBO Journal policy for the transcript of the editorial process (containing referee reports and your response letter) to be published as an online supplement to each paper. If you do NOT want this, you will need to inform the Editorial Office via email immediately. More information is available here:

<https://www.embopress.org/page/journal/14602075/authorguide#transparentprocess>

Your manuscript will be processed for publication in the journal by EMBO Press. Manuscripts in the PDF and electronic editions of The EMBO Journal will be copy edited, and you will be provided with page proofs prior to publication. Please note that supplementary information is not included in the proofs.

Please note that you will be contacted by Wiley Author Services to complete licensing and payment information. The 'Page Charges Authorization Form' is available here:

https://www.embopress.org/pb-assets/embo-site/tej_apc.pdf

Should you be planning a Press Release on your article, please get in contact with embojournal@wiley.com as early as possible, in order to coordinate publication and release dates.

If you have any questions, please do not hesitate to call or email the Editorial Office. Thank you for your contribution to The EMBO Journal.

Corresponding Author Name: Josef M. Penninger

Journal Submitted to: EMBO J

Manuscript Number: 2021-108375R